# Genome-scale metabolic modeling reveals key features of a minimal gene set

Jean-Christophe Lachance[1], Dominick Matteau[1], Joëlle Brodeur[1], Colton J Lloyd[2], Nathan Mih[2], Zachary A King[2], Thomas F Knight[3], Adam M Feist[2,4] [iD], Jonathan M Monk[2], Bernhard O Palsson[2,4,5,6] [iD], Pierre-Étienne Jacques[1] & Sébastien Rodrigue[1,*] [iD]

## Abstract

*Mesoplasma florum*, a fast-growing near-minimal organism, is a compelling model to explore rational genome designs. Using sequence and structural homology, the set of metabolic functions its genome encodes was identified, allowing the reconstruction of a metabolic network representing ~ 30% of its protein-coding genes. Growth medium simplification enabled substrate uptake and product secretion rate quantification which, along with experimental biomass composition, were integrated as species-specific constraints to produce the functional *i*JL208 genome-scale model (GEM) of metabolism. Genome-wide expression and essentiality datasets as well as growth data on various carbohydrates were used to validate and refine *i*JL208. Discrepancies between model predictions and observations were mechanistically explained using protein structures and network analysis. *i*JL208 was also used to propose an *in silico* reduced genome. Comparing this prediction to the minimal cell JCVI-syn3.0 and its parent JCVI-syn1.0 revealed key features of a minimal gene set. *i*JL208 is a stepping-stone toward model-driven whole-genome engineering.

**Keywords** genome design; genome-scale models; *Mesoplasma florum*; minimal cells; synthetic biology

**Subject Categories** Metabolism; Microbiology, Virology & Host Pathogen Interaction

**Mol Syst Biol. (2021) 17: e10099**

## Introduction

The increased efficiency of *in vitro* DNA synthesis and assembly methods (Hughes & Ellington, 2017) has enabled the development of organisms living with either large fractions or completely synthetic genomes (Gibson *et al*, 2010; Hutchison *et al*, 2016;

Venetz *et al*, 2019). The capability to physically write entire chromosomes from synthetic DNA is now an achieved ambition of synthetic genomics, but the ability to predict whether or not this assembly will produce a viable cell remains a substantial challenge. This difficulty is linked to the inherent complexity of living organisms and the incomplete knowledge of the molecular functions they entail (Danchin & Fang, 2016).

Minimal cells are simple organisms containing the fewest number of genes necessary to support self-replicating life (Glass *et al*, 2017). The number of unknown molecular functions within these small genomes is proportional to their size (Price *et al*, 2018), which makes them especially amenable to the exhaustive characterization of their content. JCVI-syn3A, a working approximation of a minimal cell, was recently reported to contain only 91 proteins of unknown function (Breuer *et al*, 2019). This number was considerably higher for the phylogenetically related *Mycoplasma pneumoniae* (311 unknowns) and even higher in the model organism *Escherichia coli* (1,780 unknowns) (Breuer *et al*, 2019).

Addressing the lack of knowledge in a given organism can be aided by a computational framework (Yurkovich & Palsson, 2016) and could lead to a complete understanding of its molecular functions, an important milestone for reliable biological engineering (Danchin & Fang, 2016; Lachance *et al*, 2019b). Furthermore, such computational frameworks can be used directly for the prediction and design of minimal gene sets (Wang & Maranas, 2018; Rees-Garbutt *et al*, 2020). The development of a genome-scale model (GEM) of metabolism, which details all known metabolic reactions catalyzed by an organism in a reaction matrix, represents a promising strategy to face this challenge (Orth *et al*, 2010; O'brien *et al*, 2013). GEMs have been previously produced for other naturally occurring or synthetic minimal cells from the Mollicutes phylogenetic group (Tomita, 2001; Suthers *et al*, 2009; Karr *et al*, 2012; Bautista *et al*, 2013; Wodke *et al*, 2013; Breuer *et al*, 2019) but not for the fast-growing and non-pathogenic *Mesoplasma florum* (see Appendix Table S1 for a summary of *M. florum* characteristics). Using this mathematically structured knowledgebase, key

1  Département de Biologie, Université de Sherbrooke, Sherbrooke, QC, Canada
2  Department of Bioengineering, University of California, San Diego, La Jolla, CA, USA
3  Ginkgo Bioworks, Boston, MA, USA
4  Department of Pediatrics, University of California, San Diego, La Jolla, CA, USA
5  Bioinformatics and Systems Biology Program, University of California, San Diego, La Jolla, CA, USA
6  Novo Nordisk Foundation Center for Biosustainability, Technical University of Denmark, Lyngby, Denmark
   *Corresponding author. Tel: +18198218000 #62939; Fax: +18198218049; E-mail: sebastien.rodrigue@usherbrooke.ca

phenotypic predictions such as gene essentiality, metabolic flux states, and growth medium requirements can be obtained from the genotype without the need for precise enzyme kinetic data (Orth *et al*, 2010; O'brien *et al*, 2013).

Here, we present *i*JL208, the first GEM for *M. florum*. The 208 genes in the model account for ~ 30% of the total gene count in the genome. We thoroughly investigated and reviewed the genome annotation using a combination of computational approaches, resulting in a metabolic network composed of 370 reactions. A recent deep characterization study of *M. florum* (Matteau *et al*, 2020) was leveraged to define a species-specific biomass composition. A novel semi-defined growth medium was developed, enabling the identification of the main energy sources that can be metabolized by *M. florum*. Both substrate uptake and product secretion rates were determined in this medium, allowing the definition of constraints on the model. Flux-state and gene essentiality predictions were validated against genome-wide expression and essentiality datasets, reaching an accuracy of ~ 78 and ~ 77%, respectively.

Finally, we took advantage of the phylogenetic proximity of *M. florum* to the minimal cell *Mycoplasma mycoides* JCVI-syn3.0 (Hutchison *et al*, 2016) to assess the predictive power of GEMs for the design of minimal genomes. We previously reported that an alternate minimal gene set was likely for *M. florum* (Baby *et al*, 2018b), which motivated our model-driven search. Given that whole-genome cloning and transplantation techniques were developed for *M. florum* (Matteau *et al*, 2017; Baby *et al*, 2018a), minimal genome designs could be put to the test imminently. This contrasts with other mycoplasmas for which predictions were made (Rees-Garbutt *et al*, 2020) but genetic engineering techniques remain unavailable. The experimentally validated *i*JL208 model was therefore used to formulate a minimal genome prediction that also accounts for both transcription unit architecture and genome-wide essentiality.

## Results

### Identification of protein molecular functions in *M. florum*

The reconstruction of a high-quality GEM for *M. florum* requires a comprehensive identification of the molecular functions encoded in its genome. We used a combination of three different computational approaches relying on both sequence and structural homology to review the annotation of all open reading frames (Appendix Supplementary Text, Datasets EV1–EV3). Proteome comparison (Fig 1A and B), structural homology (Fig 1C), and the probabilistic identification of enzyme commission (EC) numbers (Fig 1D) were combined to define a final annotation score for each of the 676 *M. florum*-predicted proteins (Fig 1E, Materials and Methods). Basic, medium, and high confidence levels could be attributed to 275, 285, and 116 proteins, respectively (Fig 1F).

### Genome-scale metabolic network reconstruction

Public databases were queried to identify the reactions associated with gene annotations included in the reconstruction (Artimo *et al*, 2012; Kanehisa *et al*, 2016; King *et al*, 2016; Placzek *et al*, 2017; Wattam *et al*, 2017). To ensure consistency between the identified reactions and available knowledge on Mollicutes metabolism, an extensive literature search was also conducted (Appendix Supplementary Text). The small size of the *M. florum* genome allowed for a manual curation of the putative function for every gene. The resulting metabolic reconstruction, *i*JL208, contains 208 protein-coding genes, 370 reactions, and 351 metabolites, a count similar to other Mollicutes models (Suthers *et al*, 2009; Bautista *et al*, 2013; Wodke *et al*, 2013; Breuer *et al*, 2019) (Fig 2, Table 1, Dataset EV4, Codes EV1 and EV2).

Overall, 236 of the 370 reactions are gene-associated in *i*JL208 (Fig 2 and Dataset EV4). Of those, 156 reactions are linked to a single gene while 80 are linked to more than one (enzyme complex or isozymes). Of the 134 orphan reactions, 93 are pseudo-reactions (85 extracellular exchanges, three intracellular sinks, one ATP maintenance, and four biomass reactions) while 41 are necessary orphans (15 spontaneous and 26 orphan transport reactions). Notably, about a third (84/277) of the total number of reactions (excluding pseudo-reactions) in the model are transport reactions.

*i*JL208 reactions were grouped into six different modules: Energy, Amino acids, Lipids, Glycans, Nucleotides, and Vitamins & Cofactors (Fig 2). An extensive description of the composition of each module and the mechanistic description of all reactions included in the model is provided in the Appendix Supplementary Text and Dataset EV4. Each module describes a general metabolic objective and contains between 15 (Glycans) and 76 (Nucleotides) reactions, as well as 14 (Glycans) to 57 (Energy) genes.

The extent of missing knowledge in each module was estimated using the number of orphan reactions required to generate a

---

**Figure 1. Computational identification of molecular functions in *Mesoplasma florum*.**

An example of the characterization process is provided with the enolase gene.

A   Predicted amino acid sequence of the *M. florum* enolase (Mfl468).

B   The PATRIC proteome comparison tool allowed the identification of orthologs in the four Mollicutes species for which a metabolic model was available, including the reactions to which they were associated in other models as well as their gene names and/or enzyme commission (EC) numbers.

C   The Structural Systems Biology software (ssbio) was used to search for known protein domains in the amino acid sequence of proteins using the Protein Data Bank (PDB) as a reference. If any domain was detected, the I-TASSER suite was used to generate a tridimensional model of the protein, which was in turn used to obtain an EC number prediction with COFACTOR.

D   Gold standard EC number identifications (yellow bar) were found using DETECT v2, which provides the likelihood of correct annotation (top). These predictions were compared with those from RefSeq, PATRIC, and COFACTOR (bottom).

E   Sankey diagram presenting the sequential steps used to interrogate the 676 predicted coding sequences. The level of confidence specific to each approach was determined and is represented by the red (low) to blue (high) color gradient.

F   Distribution of the final annotation score for the 676 predicted *M. florum* protein-coding genes. Based on this score, a basic (< 3; red), medium (≥ 3 and < 7; kaki), or high (≥ 7; green) confidence level could be attributed to each predicted protein.

---

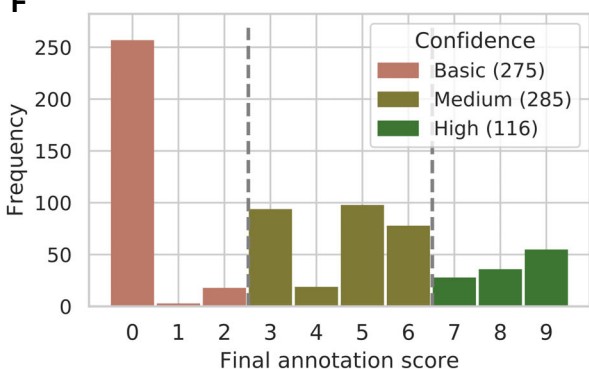

**Figure 1.**

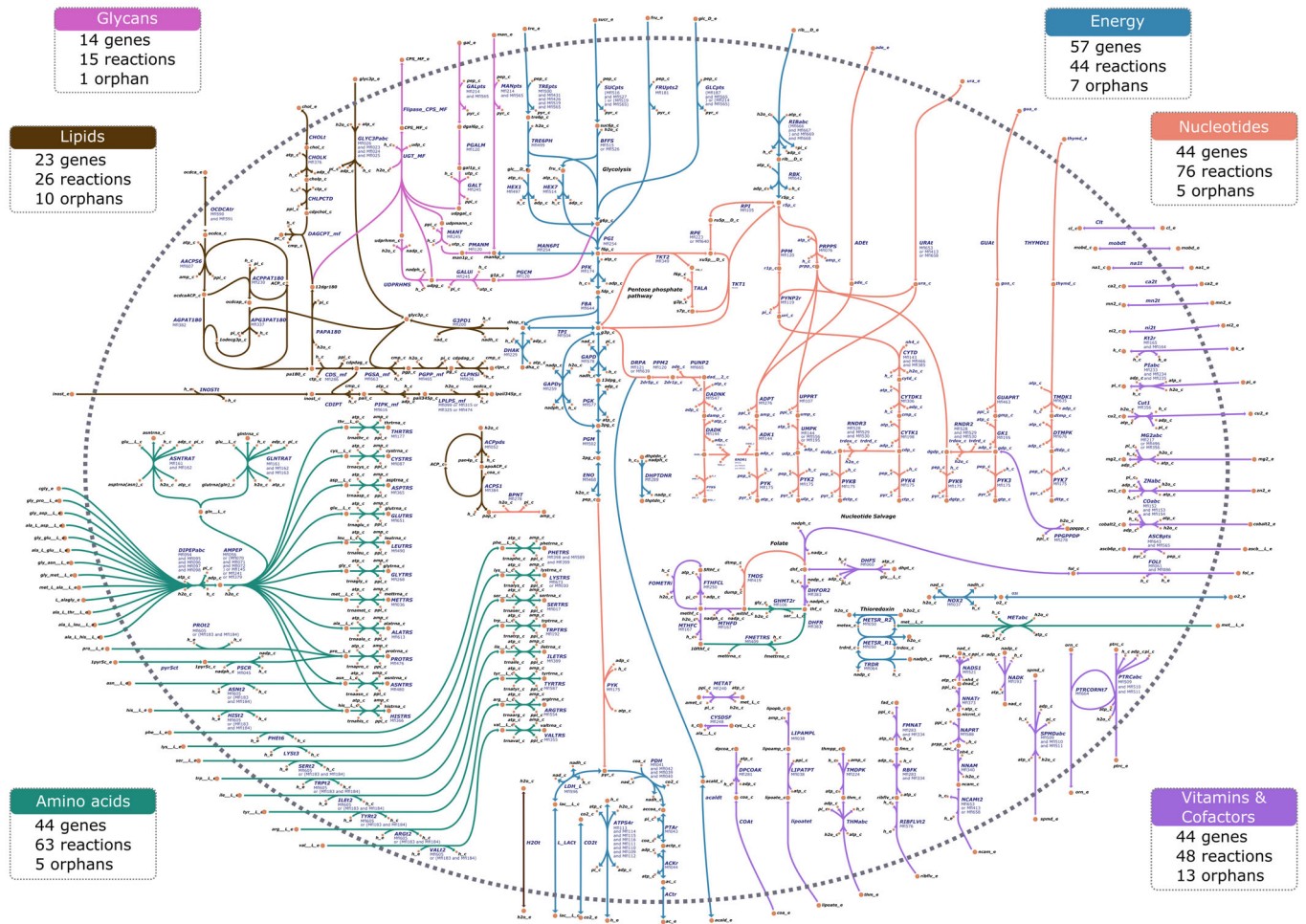

**Figure 2. Map of the genome-scale metabolic network of *Mesoplasma florum*.**

Circles represent metabolites, and connecting lines indicate metabolic reactions. Metabolite names are indicated in black, while reaction names and associated gene names are in dark blue. Reaction directionality is represented by arrows. The dotted line shows the cell membrane, with transport reactions linking the intracellular milieu with the extracellular environment. The reactions are color-coded according to the six main modules. The outer colored boxes describe the number of genes, gene-associated reactions, and orphan reactions. An interactive Escher (King *et al*, 2015) version of this map is available (Code EV1).

**Table 1. Number of protein-coding genes, reactions, and metabolites in Mollicutes metabolic models. The number of genes and reactions shared with *Mesoplasma florum* is represented for each model.**

| Species | Model | Genes: Model/Total (%) | Total reactions in model | Reactions: shared with *M. florum* (%)[a] | Total metabolites in model |
|---|---|---|---|---|---|
| *Mycoplasma genitalium* | *i*PS189 | 126/507 (24.9%) | 351 | 79/174 (45.4%) | 324 |
| *Mycoplasma pneumoniae* | *i*JW145 | 145/691 (20.1%) | 306 | 74/156 (47.4%) | 346 |
| *Mycoplasma gallisepticum* | N/A[b] | 198/747 (26.5%) | 322 | 83/260 (31.9%) | 444 |
| JCVI-syn3A | N/A[b] | 155/473 (32.8%) | 338 | 87/338 (25.7%) | 304 |
| *M. florum* | *i*JL208 | 208/680 (30.6%) | 370 | — | 351 |

[a]The percentage of shared reactions applies only to the gene-associated reactions.
[b]No model name was provided by the authors.

functional model. Energy, Nucleotides, and Amino acids modules had the lowest proportion of orphan reactions (< 14%; Fig 3A) and describe well-known aspects of the metabolism of Mollicutes.

Conversely, the Lipids and the Vitamins & Cofactors modules had the highest percentage of orphans relative to their total number of associated reactions (25 and 33%, respectively). While the Glycans

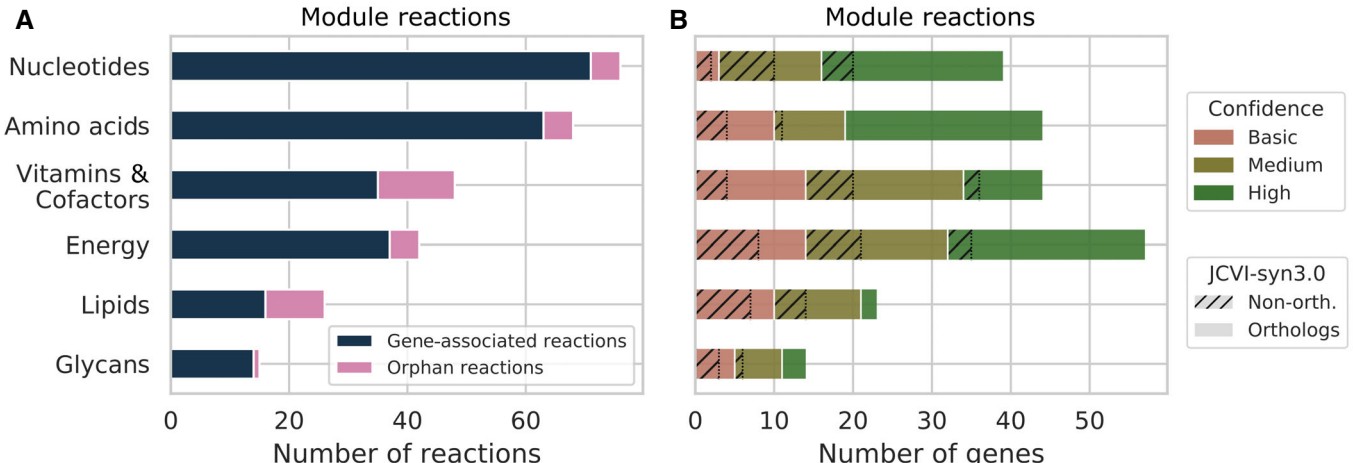

**Figure 3. Characteristics of the *Mesoplasma florum* metabolism as revealed by the genome-scale network reconstruction.**

A  Distribution of gene-associated and orphan reactions in the six modules defined in *i*JL208.

B  Distribution of genes and their associated final annotation score (see Fig 1F) in the metabolic modules. JCVI-syn3.0 orthologs are plain while the non-orthologs are hatched.

module contains a single orphan reaction, the majority of its reactions (12/15) are assumed promiscuous reactions. In proportion to their total number of genes, these three modules also displayed a lower gene annotation confidence level than the other three (Fig 3B).

As expected, about 70% (146/208) of the protein-coding genes included in the metabolic reconstruction were orthologous to JCVI-syn3.0, a slightly higher fraction than the number of orthologs present in the entire genome (~ 60%, 411/680; Fig 3B and Appendix Fig S1). The Glycans and Amino acids modules were more conserved than the average for the entire model, while Energy, Vitamins & Cofactors, and Nucleotides had a distribution similar to the model. The Lipids module was the least conserved with 52% of orthologs (Fig 3B).

**Medium simplification and growth kinetics**

While the genomic complexity of Mollicutes is remarkably low (Sirand-Pugnet *et al*, 2007), the number of necessary medium components to sustain their growth is rather high (Keçeli & Miles, 2002). The metabolic reconstruction reflects this reality with 84 extracellular metabolites associated with transport reactions (Fig 2 and Dataset EV4). We sought to elaborate a simplified growth medium for *M. florum* which, before this study, was commonly grown in the complex and undefined ATCC 1161 medium containing horse serum (HS), yeast extract (YE), and heart infusion broth (see Materials and Methods).

A particular problem faced when using the ATCC 1161 medium was apparent growth when no sucrose was added (Fig 4A, top), which prevented the assessment of *M. florum*'s metabolic capabilities when supplemented with different carbohydrates. To circumvent this issue, the concentrations of HS and YE were lowered by adding a completely defined rich medium base to the mixture (CMRL-1066). This allowed a 64-fold reduction in the concentrations of HS (to 0.313%) and YE (to 0.02%) required for significant

growth (Appendix Fig S3), as well as the complete removal of heart infusion broth. This CMRL-1066-based semi-defined medium, referred to as CSY, allows visible growth only when sucrose is added (Fig 4A, bottom).

We observed that reducing the concentration of HS and YE impacted the doubling time of *M. florum* (Fig 4B), suggesting that nutrients contained in these undefined components are rate-limiting in *M. florum*. When varying the initial sucrose concentrations in CSY medium, the total biomass produced followed an asymptotic behavior (Fig 4C), with a predicted maximum concentration of 5.95 x 1e8 colony-forming unit per ml (CFU/ml), corresponding to 0.013 grams of dry weight per liter (gDW/l). Given that *M. florum* cell densities typically reach ~ 1e10 in ATCC 1161 medium (Matteau *et al*, 2020), these observations confirmed that nutrients other than sucrose are rate-limiting in CSY.

Nevertheless, at low sucrose concentrations, initial sucrose and growth rate displayed a Monod-like relation (Monod, 1949), with a maximal growth rate in CSY found at 0.44 h$^{-1}$ (Fig 4D, Appendix Figs S4 and S5A). We used this range of dependency between the growth rate and the initial sucrose concentrations to define substrate uptake and by-product secretion rates. The sucrose and combined lactate/acetate variation of concentration over time was measured by high-performance liquid chromatography (HPLC). Substrate-specific uptake rate (sucrose) and metabolic by-product secretion rates (lactate/acetate) were calculated using linear regression in the exponential growth phase (Materials and Methods; Appendix Figs S5 and S6). The range of possible rates within the exponential phase was calculated for each initial sucrose concentration (Fig 4E and F). Interestingly, a tendency toward both a maximum uptake and secretion rates ($qS_{max}$) could be observed, which is desired for modeling purposes where the optimal conditions are assumed. Accounting for experimental variability and the number of data points available, the maximum sucrose uptake rate was set at −5.26 mmol/gDW/h and the combined lactate/acetate secretion rate at 8.69 mmol/gDW/h (see Materials and Methods).

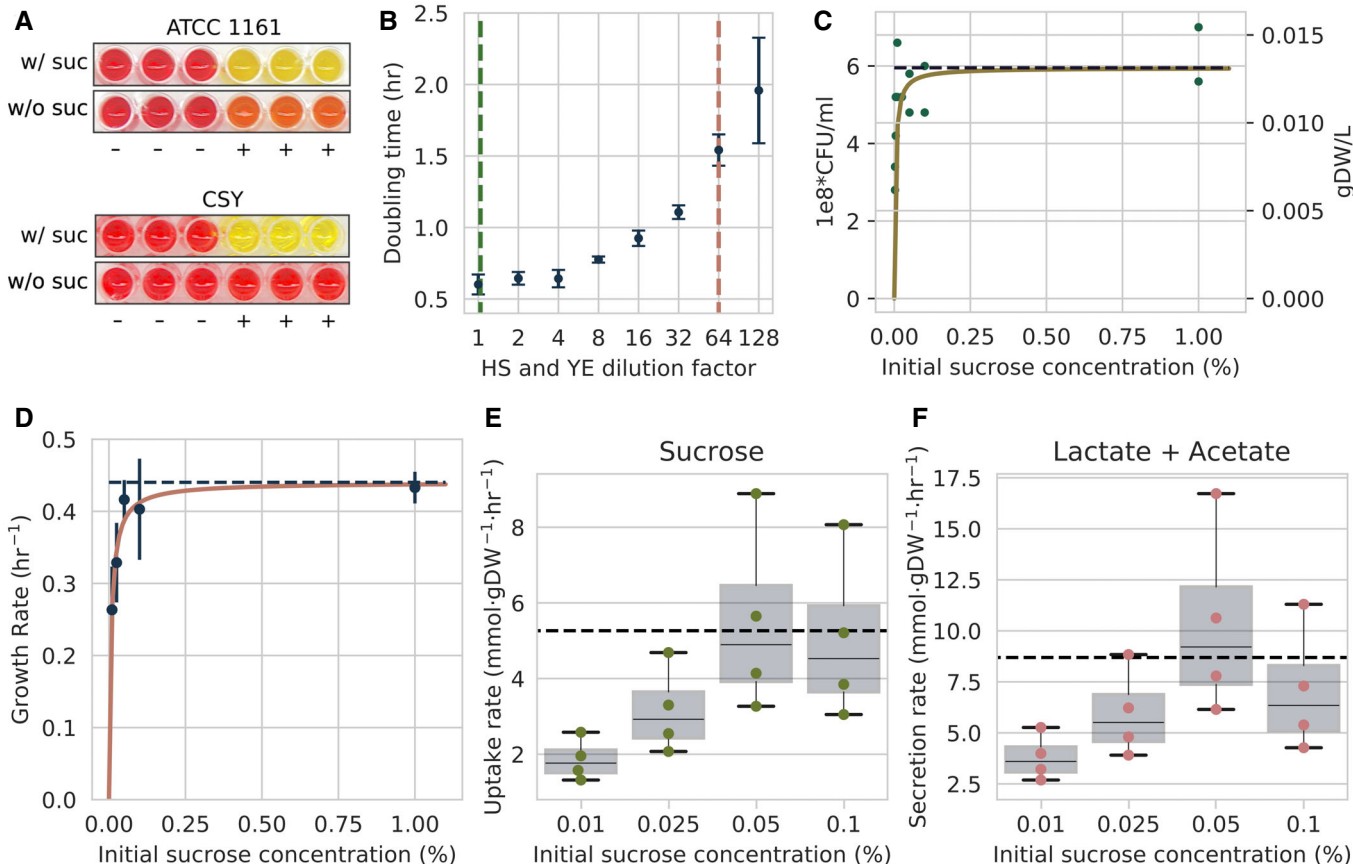

**Figure 4. Impact of medium composition on *Mesoplasma florum* growth kinetics.**

A   Bacterial growth assessed by color change due to medium acidification for both the undefined ATCC 1161 and semi-defined CSY media with or without sucrose, 4% and 1%, respectively. (+), inoculated wells; (−), non-inoculated control. The picture was taken after a 24-h incubation period.

B   Impact of horse serum (HS) and yeast extract (YE) dilution on *M. florum* doubling time. The green and red dotted lines indicate HS and YE concentrations found in ATCC 1161 medium (20% HS, 1.35% YE) and CSY medium (0.313% HS, 0.02% YE), respectively. Doubling times were measured in a CMRL-1066 base medium using colorimetric assays. Dots and error bars indicate the mean and standard deviation calculated from three biological replicates.

C   The maximal biomass concentration observed for *M. florum* cultures growing in CSY medium with varying initial concentrations of sucrose. Biomass was measured using colony-forming units (CFU/ml; left axis) and converted to grams of dry weight (gDW/l; right axis). A rectangular hyperbola fit is shown (yellow), and the dotted line represents the maximal biomass value predicted by the fit (1e8 × 5.95 CFU/ml; 0.013 gDW/l).

D   Relationship between varying initial sucrose concentration and growth rate in CSY medium. Growth rates were determined by fitting a simple exponential growth model to CFU/ml data from time-course experiments, and error bars indicate the standard deviation associated with each value. CFU/ml quantifications were performed in technical duplicate as described in Appendix Figs S4 and S5. A rectangular hyperbola fit is shown, and the predicted maximal growth rate (0.44 h$^{-1}$) is indicated by the dotted line.

E, F   Sucrose-specific uptake rate (E) and combined lactate/acetate-specific secretion rates (F) at varying initial sucrose concentrations in CSY medium. Boxplots represent the median and interquartile range of uptake or secretion rate values calculated at different time intervals during the exponential growth phase of *M. florum* cultures. Whiskers indicate minimal and maximal values. Dotted lines indicate the selected uptake (−5.26 mmol/gDW/h) and secretion rate (8.69 mmol/gDW/h) values used for modeling. Sucrose quantifications were performed in technical duplicate, whereas lactate and acetate quantifications were performed in single replicates. See Appendix Fig S5 for further details.

## Conversion into a mathematical format and sensitivity analysis

The biomass objective function (BOF) is a reaction of the stoichiometric matrix used to simulate an organism's growth (Feist & Palsson, 2010). Previously reported experimental macromolecular composition characterizing 98.8% of *M. florum*'s dry mass, as well as multiple omics datasets (Matteau *et al*, 2020) were used as input into the BOFdat software (Lachance *et al*, 2019a) to define the *M. florum*-specific BOF (Materials and Methods; Appendix Supplementary Text). DNA, RNA, and protein stoichiometric coefficients were

determined by the first step of BOFdat and accounted for 76.4% of the total cellular dry weight (Matteau *et al*, 2020) (Fig 5A, left). Coenzymes and inorganic ions were next identified, finding 12 metabolites previously defined as universally essential cofactors in prokaryotes (Xavier *et al*, 2017), as well as seven other metabolites with high connectivities (Fig 5A, middle). These 19 metabolites were considered as the soluble pool, and their stoichiometric coefficients were determined using the remaining 1.2% *M. florum* biomass.

Using the Step 3 of BOFdat, the correspondence between single-gene essentiality prediction and genome-wide transposon

mutagenesis data (Baby *et al*, 2018b) was improved by the addition of nine metabolites, two of which were also identified in previously published lipidomic data (Matteau *et al*, 2020) (Fig 5A, right; Table 2, Appendix Fig S7, Materials and Methods). A metabolite corresponding to the *M. florum* capsular polysaccharide (CPS) was also added during Step 3. In the pathways leading to the production of the four most frequently identified metabolites in Step 3, 10 out of 15 genes had their essentiality prediction modified compared with Step 2 (Fig 5B). Among these 10, a single gene (*mfl061*) was wrongfully identified as essential compared with transposon mutagenesis data. The final *iJL208* biomass composition is presented in Appendix Table S3.

The model was then constrained using the experimental rates defined above (Fig 4D–F). Given the complexity of the growth medium, growth- and non-growth-associated maintenance costs (Varma & Palsson, 1993) (GAM and NGAM, respectively) could not be obtained directly and had to be inferred from known parameters. To simulate growth on CSY, the *in silico* minimal medium was defined using the COBRApy toolbox (Ebrahim *et al*, 2013) (Appendix Table S2) and a set of key initial parameters were selected (Materials and Methods). A phenotypic phase plane analysis (Edwards *et al*, 2002) was performed to identify the ATP maintenance value that allowed the model to reproduce the experimentally determined growth rate and define both GAM (Fig 5C) and NGAM (Fig 5D) values. With these constraints, the model sensitivity to the sucrose uptake rate was assessed, revealing three different growth phases ending with a plateau (Fig 5E). This is similar to the experimental observations that, in CSY, *M. florum*'s growth is not restricted by sucrose availability alone (Fig 4). The main model constraints identified in this study are listed in Table 3.

The metabolic reconstruction revealed the capability of *M. florum* to produce both lactate and acetate as fermentation products (Fig 2). Since only their combined secretion could be measured by HPLC (Appendix Fig S5C, Materials and Methods), individual production rates had to be inferred. Previously reported differential expression of key enzymes (Matteau *et al*, 2020), such as the lactate dehydrogenase (LDH: Mfl596), showed an approximately 4- to 8-fold increase in the protein expression levels compared with genes of the pyruvate dehydrogenase complex (PDH: Mfl039, Mfl040, Mfl041, and Mfl042). We thus used an 8:1 initial ratio (lactate:acetate) for further phenotypic phase plane analysis (Materials and Methods). For lactate production, this analysis revealed a positive linear relationship between the predicted growth rate and lactate secretion rate (Fig 5F). Conversely, the production of acetate was detrimental to the predicted growth rate (Fig 5G), and its secretion rate had to be lowered to match the experimental growth rate in CSY. While both secretion routes generate ATP, acetate production requires oxygen to regenerate the $NAD^+$ pool through the NADH oxidase (NOX2, Mfl037; Fig 2). To simultaneously ensure that *M. florum* was not able to produce oxygen and that the growth rate was not linearly dependent on oxygen uptake, the NOX2 reaction was bounded between 0 and 5 mmol/gDW/h, resulting in an optimal oxygen uptake rate intersecting a plateau (Fig 5H).

**Validation of model phenotypic predictions**

The development of the CSY medium enabled an experimental validation of the capability of *M. florum* to grow on 14 different carbohydrates and to compare these observations with the constrained model's phenotypic predictions (Figs 6A and EV1 and Appendix Supplementary Text). The growth/no-growth phenotype was correctly predicted by *iJL208* for 12 out of the 14 sugars tested. The two remaining sugars, maltose and mannose, were used by *M. florum* while the model predicted no growth. To address these discrepancies, the alternate carbon metabolism of *M. florum* was studied, seeking enzymes that would likely carry a promiscuous activity. Particularly, the specificity of three enzymes was challenged using the FATCAT 2.0 server (Li *et al*, 2020) to compare the tridimensional structures reconstructed with I-TASSER in this study to crystallographic structures from the Protein Data Bank (PDB; Figs 6B and EV2, and Appendix Table S4).

The similarity between maltose and trehalose suggested that the trehalose hydrolase (Mfl499) could also hydrolyze maltose. This hypothesis was supported by the very high structural similarity between the reconstructed Mfl499 and a *Bacillus* sp. α-glucosidase shown to have a high specificity for α-(1-4)-glucosidic linkage (Auie-wiriyanukul *et al*, 2018) (Figs 6B(i) and EV2A). Similarly, the capacity of *M. florum* to metabolize mannose could be explained by the capability of the glucose-6-phosphate isomerase (Mfl254) to convert mannose-6-phosphate into fructose-6-phosphate, hereby entering glycolysis (Figs 6B(ii) and EV2B). While the promiscuity of phosphotransferase systems (PTS) and other transporter complexes could not be tested *in silico* (Materials and Methods), the addition of both the promiscuous transport and digestion reactions was sufficient to provide a growth prediction on maltose and mannose. As reported previously, both glucose and mannose were detected in the *M. florum* polysaccharide layer (Matteau *et al*, 2020). The presence of a phosphomannomutase (Mfl120) in the genome annotation suggested the conversion of mannose-6-phosphate to mannose-1-phosphate, a necessary precursor for glycan synthesis (Bertin *et al*, 2015). The very high structural similarity between the reconstructed Mfl120 and an enzyme necessary for the production of exopolysaccharides (Regni *et al*, 2002) in *Pseudomonas aeruginosa* (Figs 6B(iii) and EV2C) supported this hypothesis.

Genome-wide expression (Matteau *et al*, 2020) and transposon mutagenesis (Baby *et al*, 2018b) datasets available for *M. florum* were used as a reference for the validation of model flux states and gene essentiality predictions (Thiele & Palsson, 2010). From the 208 protein-coding genes present in *iJL208*, 173 showed a consistent expression between transcriptomic and proteomic datasets (Appendix Fig S8, Materials and Methods, and Dataset EV5). Of these genes, 135 occurrences had an expression profile in agreement with the metabolic fluxes predicted by the model, which corresponds to an accuracy of 78.03% (Fig 6C). In parallel, the original transposon mutagenesis data were re-analyzed following the method proposed by Hutchison and colleagues (Hutchison *et al*, 2016), where genes hit solely in the final 20% of their nucleotide sequence were not considered as essential. Using this approach resulted in the re-assignment of 79 coding genes previously considered as non-essential, for a total of 332 *M. florum* genes now determined as essential (Fig EV3, Dataset EV5, and Materials and Methods). 160 single-gene essentiality predictions from *iJL208* were consistent with that revised experimental data, for an overall accuracy of 76.92% (Fig 6D).

Essentiality and expression comparison with model predictions provided a context for the refinement of the model. Targeted false

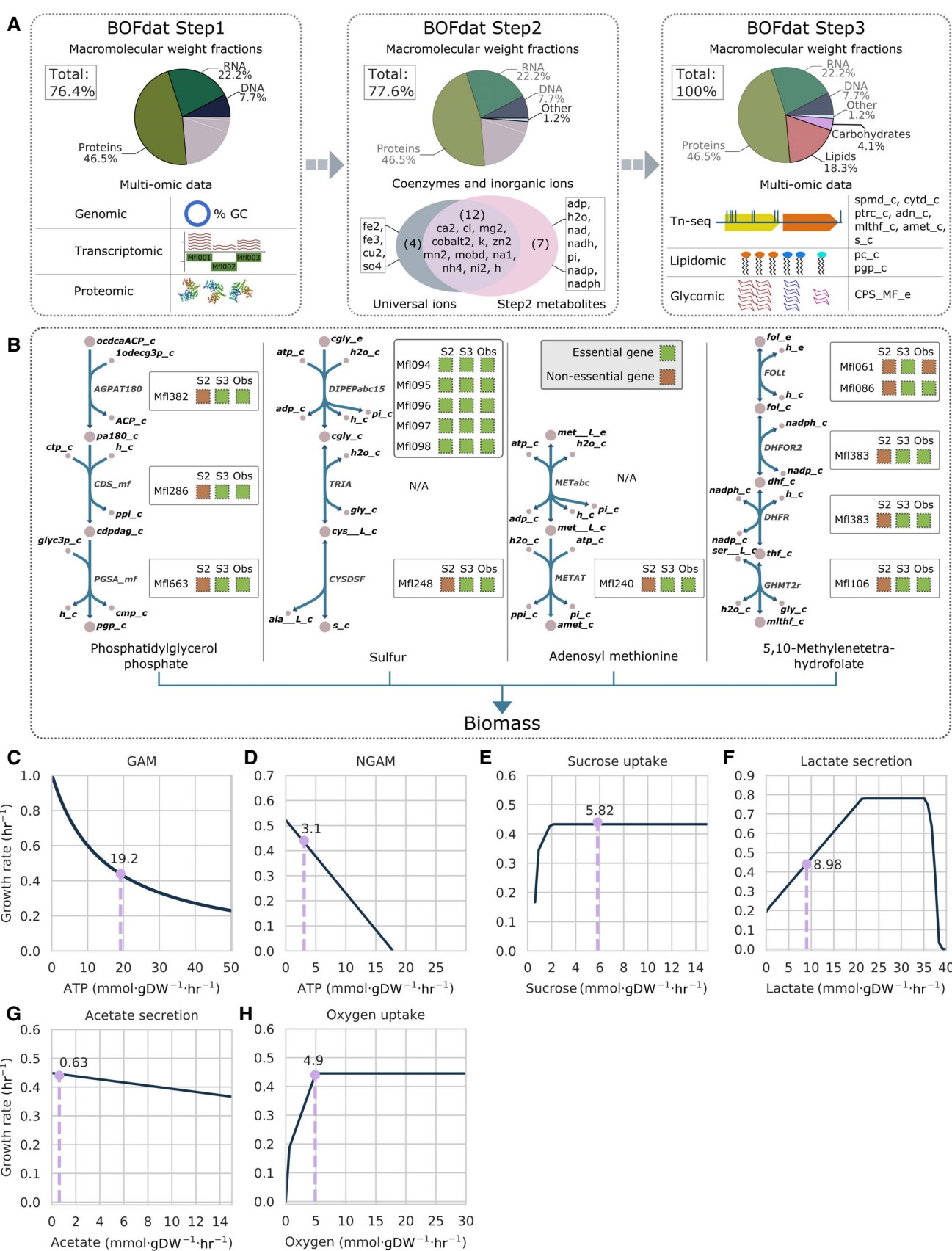

**Figure 5.**

**Figure 5. Conversion into a mathematical format.**

A    The biomass objective function (BOF) was defined using the BOFdat software (Lachance *et al*, 2019a) and experimental data from Matteau *et al* (2020). The stoichiometric coefficients for the major cellular macromolecules were generated with BOFdat Step 1 (left), the inorganic ions and coenzymes with BOFdat Step 2 (middle), and the remaining components with BOFdat Step 3 (right), which identifies the metabolites with the greatest impact on the BOF gene essentiality prediction accuracy.

B    Depiction of the metabolic context of four complete pathways supporting the output of BOFdat Step 3. Metabolite names are shown in black and reaction names in gray. Boxes contain the predicted essentiality of the protein-coding genes associated with a given reaction following BOFdat Step 2 (S2) and Step 3 (S3), along with the observed essentiality (Obs). N/A corresponds to orphan reactions.

C–H  Sensitivity analysis of the *i*JL208 model to (C) growth-associated maintenance (GAM), (D) non-growth-associated maintenance (NGAM), (E) sucrose uptake rate, (F) lactate and (G) acetate secretion rates, and (H) oxygen uptake rate. The light purple dotted line represents the corresponding rate at the predicted maximal growth rate in CSY (0.44 h$^{-1}$).

**Table 2. Comparison of the metabolites identified in BOFdat Step 3 to those included in other Mollicutes' model biomass compositions.**

| Metabolite | *Mycoplasma genitalium* | *Mycoplasma pneumoniae* | *Mycoplasma gallisepticum* | JCVI-syn3A |
|---|---|---|---|---|
| Sulfur | Present[a] | Absent | Present[a] | Absent |
| Spermidine | Present | Absent | Present | Present |
| Cytidine | Absent | Present | Absent | Absent |
| Putrescine | Present | Absent | Present | Absent |
| Phosphatidylglycerol | Absent | Absent | Present | Present |
| Adenosine | Absent | Present | Absent | Absent |
| Methyltetrahydrofolate | Present | Absent | Present | Present |
| S-adenosyl-methionine | Present | Present | Present | Absent |
| Phosphatidyl-glycerophosphate | Absent | Absent | Absent | Absent |

[a]Sulfate was identified instead of Sulfur in these models.

**Table 3. Comparison of the main model constraints with those of other Mollicutes models. All units are given in mmol/gDW/h.**

| Constraint | *Mesoplasma florum* (this study) | *Mycoplasma genitalium* (Suthers *et al*, 2009; Karr *et al*, 2012) | *Mycoplasma pneumoniae* (Wodke *et al*, 2013) | *Mycoplasma gallisepticum* (Bautista *et al*, 2013) | JCVI-syn3A (Breuer *et al*, 2019) |
|---|---|---|---|---|---|
| GAM | 17.2 | 9.7[a] | 25 | 9.7[b] | 46.54[b] |
| NGAM | 3.1 | 8.4[a] | 3.3 | 8.4[b] | 3.3[b] |
| Substrate uptake rate | 5.26 | 5[a] | 7.37 | 16.53 | 7.4[b] |
| Acetate secretion rate | 0.53 | Unconstrained | 6.93 | N/A | 6.9[b] |
| Lactate secretion rate | 8.16 | Unconstrained | N/A | 10.29 | Unconstrained |

[a]Extrapolated from other species.
[b]Extrapolated from other Mollicutes models.

negatives, i.e., genes simultaneously expressed and essential while no flux or essentiality was predicted in *i*JL208 (Fig 6C and D), together with a single false positive was manually curated by the addition of specific constraint(s) (Appendix Supplementary Text and Appendix Fig S9). Among those, genes of the pentose phosphate pathway (PPP) were specifically investigated since they all showed expression but carried no metabolic flux (Fig 6E). In *M. florum* and other Mollicutes, this pathway is incomplete because no gene is typically attributed to the transaldolase (TALA) reaction (Miles, 1992; Suthers *et al*, 2009; Wodke *et al*, 2013; Breuer *et al*, 2019). Here, the I-TASSER reconstructed structure of two 2-deoxyribose-5-phosphate aldolases (Mfl121 and Mfl639) were queried against the PDB to find potential matches with transaldolase structures (Fig EV2D–F). Of the 12 transaldolases identified, the structure from *Thermotoga maritima* had the most significant match

and highest similarity (Fig 6E and Appendix Table S4). While this structural similarity points to a potential transaldolase reaction, experimental validation will be required to confirm its presence. Meanwhile, the TALA reaction was assigned to Mfl121 or Mfl639 in *i*JL208, thereby allowing flux through the PPP, which is consistent with expression data.

**Model-driven prediction of a minimal genome**

The validated *i*JL208 GEM was used together with experimental gene essentiality and transcription unit architecture (Matteau *et al*, 2020) to infer and characterize a minimal gene set for *M. florum* using the MinGenome algorithm (Wang & Maranas, 2018). To generate this prediction, MinGenome incorporates both experimental data and model constraints into an optimization problem that

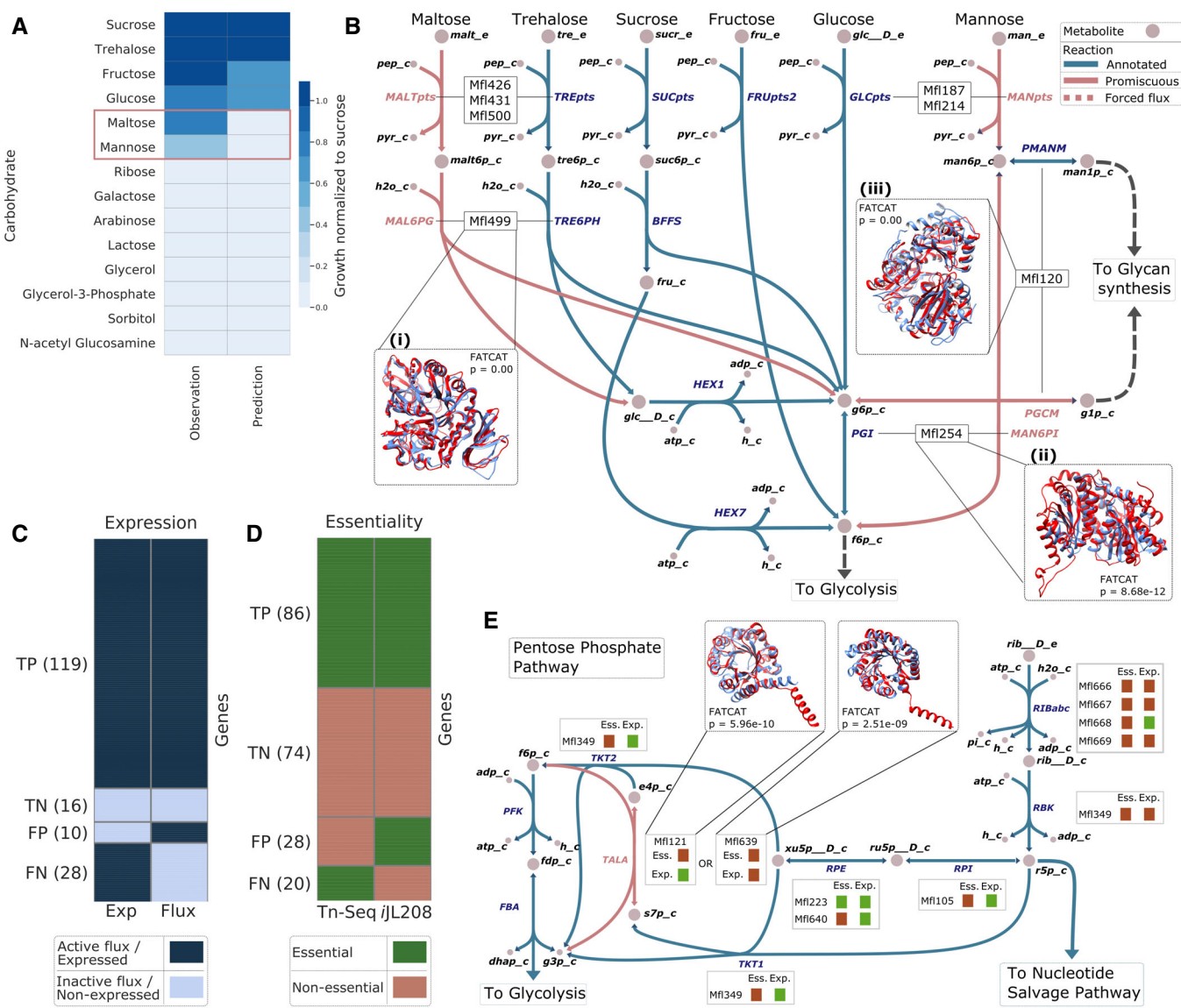

**Figure 6. Validation of the model's phenotypic predictions.**

A Growth phenotype observed on CSY medium supplemented with different carbohydrates (1% final concentration) compared with *i*JL208 predictions. Growth observations and predictions were normalized by growth on sucrose. Discrepancies between experimental data and predictions are highlighted (red rectangle).

B The metabolic network reconstruction allowed the identification of potential candidates carrying promiscuous reactions (pink) responsible for maltose and mannose catabolism. 3D structures of candidates (Mfl499, Mfl120, and Mfl254; red) are superimposed to available structures in the Protein Data Bank (light blue) for which the suspected enzymatic activity is annotated. FATCAT *P*-values are shown.

C The 173 genes where the transcriptomic and proteomic expression data (Exp) are consistent (see Appendix Fig S8) are compared with flux state predictions generated by parsimonious flux-balance analysis (Flux). True positives (TP) and true negatives (TN) correspond to genes where both predictions and observations are consistent, while false positives (FP) and false negatives (TN) where they are inconsistent. Genes considered expressed or with an active flux are represented in dark blue, and silent in light blue.

D Revised gene essentiality data (see Fig EV3) compared with essentiality predictions from *i*JL208. Essential and non-essential protein-coding genes are represented in green and red, respectively. TP, TN, FP, and TN are as in (C).

E The pentose phosphate pathway (PPP) of *Mesoplasma florum* along with ribose import. The transaldolase (TALA) reaction (pink) was missing in the network. Structural similarity between two 2-deoxyribose-phosphate aldolases (Mfl121, red in left box; Mfl639, red in right box) to a known transaldolase from the PDB (light blue) served as basis for the addition of this gene-associated reaction in the model. The observed essentiality (Ess.) and expression (Exp.) of genes (see panels (C) and (D)) associated with each reaction of the PPP are indicated by green (essential or expressed) or red (non-essential or non-expressed) colored squares.

finds the largest possible deletion in the genome which, applied iteratively, defines the minimal gene set. Interestingly, a minimum growth rate can be imposed as a constraint on the MinGenome optimization problem. We investigated the impact of a range of imposed growth rates on predicted genome reduction scenarios, and three possible genome reduction scenarios were obtained

(Appendix Supplementary Text, Dataset EV6, and Appendix Fig S10). The smallest genome size (562 kbp) was identified at the lowest imposed growth rate, corresponding to 563 retained and 152 deleted genes (535 and 145 protein-coding genes, respectively; Fig 7 A). The resulting genome designs were compared with the gene content of JCVI-syn3.0 (Hutchison *et al*, 2016), which has a high proportion of orthologs in *M. florum* (Appendix Fig S1) and provides a compelling validation framework for a minimal genome. Interestingly, lowering the growth rate constraint increased the similarity of the predicted minimal gene set to JCVI-syn3.0 (Dataset EV6 and Appendix Fig S10).

The smallest predicted *M. florum* genome was further analyzed by comparing its 535 retained and 145 deleted protein-coding genes to both JCVI-syn3.0 and its parent strain JCVI-syn1.0 (Gibson *et al*, 2010). Of the 145 proteins deleted in this scenario, 37 are present in JCVI-syn3.0, while 32 are JCVI-syn1.0 proteins that were also deleted in the process of generating JCVI-syn3.0 (Fig 7B). In total, 108 (74%) of these proteins were not in JCVI-syn3.0 (Fig 7B). Also, the number of JCVI-syn3.0 proteins with no ortholog in *M. florum* (58) was similar to the number of JCVI-syn1.0 proteins deleted to generate JCVI-syn3.0 and the number of proteins shared with *M. florum* (63).

We further investigated the functional categories of the deleted proteins and combined this information with the reported protein expression level, revealing that eight key cellular functions were simply never hit by any deletions: ATP synthase, Translation factors, RNA polymerase, Protein export, Cofactor biosynthesis, Sulfur relay system, Lipid metabolism, and the PPP metabolism (Figs 7C and EV4A, and Dataset EV6). The Ribosome category was nearly untouched except for the highly expressed ribosomal protein L31 (*rpmE*, Mfl648). While functionally important, this protein was categorized as non-essential in our dataset (Dataset EV5). In *E. coli*, it was also shown that the deletion of both this protein and its paralog would yield a viable cell, albeit one with significant growth defects (Lilleorg *et al*, 2017). Like in *M. florum*, this protein was hit by transposons in JCVI-syn1.0 and caused only minor growth disadvantages (Hutchison *et al*, 2016). However, the *rpmE* gene was retained in JCVI-syn3.0, likely to increase the robustness of the cell.

The key features retained in the proposed minimal gene set were similarly detailed by mapping their functions to KEGG categories and their homology to JCVI-syn3.0 or JCVI-syn1.0 (Figs 7D and EV4B, Dataset EV6, and Appendix Supplementary Text). The 535 proteins retained in the minimal gene set were similarly distributed between the three most represented functional categories. Of the 191 proteins that did not map to a KEGG category, the majority (106) were specific to *M. florum* (Fig EV4B). Three of the 12 sub-categories contained in the Metabolism category (ATP synthase, Amino acid metabolism, and Secretion system) were exclusively composed of proteins having orthologs in JCVI-syn3.0 (Fig 7D and Dataset EV6). While all glycolysis enzymes were retained and common with JCVI-syn3.0, enzymes responsible for the assimilation of sucrose in the three sub-categories where they were found (Transport, PTS, and Glycolysis and carbohydrate metabolism) had no orthologs in JCVI-syn3.0, meaning that energy sources are interchangeable in Mollicutes' minimal genomes.

We also observed the absence of the E1 component of the PDH complex in JCVI-syn3.0 (Hutchison *et al*, 2016). The conservation of these proteins in our minimal genome prediction relies on the forced

secretion of acetate in *i*JL208 (Fig 5), which best represented the experimental setting. Consistent with our observation that varying the imposed growth rate generated three different genome reduction scenarios, this reveals that alternate minimal genome designs may be obtained by varying the constraints imposed on the input GEM.

Finally, the retained proteins in the genetic information processing category are regrouped in 13 sub-categories and the vast majority have orthologs in JCVI-syn3.0 (174/195; Figs 7D and EV4B, and Dataset EV6). The fewest number of JCVI-syn3.0 orthologs was found in the DNA repair sub-category. The conserved proteins in our prediction entailed recombination proteins, glycosylases, and the DNA polymerase IV. In all cases, a single occurrence was still available in JCVI-syn3.0, meaning that these genome integrity functions should remain in a minimal gene set. Transcription factors involved in the assimilation of fructose were also conserved in our prediction while being absent from JCVI-syn3.0, which is consistent with the observations made for the metabolism that revealed energy sources could be swapped in minimal genomes.

## Discussion

Computer-aided design is crucial for the development of synthetic biology on a large scale. In genome-writing projects (Ostrov *et al*, 2019), the predictive power of GEMs could be leveraged to reduce the overall design and engineering efforts required to produce a viable strain. Still, the applicability of computational models for genome design is tightly linked to the level of knowledge available for the organism of interest.

In *M. florum*, the level of knowledge was examined by cross-validating the identification of molecular functions from different computational methods, establishing a confidence hierarchy for protein annotation (Fig 1, Datasets EV1–EV3). Using predicted protein functions, we reconstructed the metabolic network of *M. florum* and produced the first GEM for this near-minimal organism (Codes EV1 and EV2). Overall, *i*JL208 shares many similarities to previously reconstructed Mollicutes models (Table 1), including the requirement of a rich medium as a key feature to support the cell's growth, typically associated with a scavenger lifestyle (Arraes *et al*, 2007; Fisunov *et al*, 2016). This metabolic regime is mainly characterized by the abundance of transport reactions (84), the absence of a respiratory system, and the fact that biosynthesis occurs mostly through salvage pathways (Fig 2 and Dataset EV4). These elements are closely linked, with glycolysis providing a low ATP yield that is nonetheless sufficient to fuel the import of nutrients from the medium via various PTS. *M. florum* then assembles premade molecular building blocks, which considerably lowers the energetic cost of building cellular biomass.

Combining this information with the six modules proposed in the metabolic reconstruction (Fig 2) revealed that the Lipids, Glycans, and the Vitamins & Cofactors modules had fewer genes identified with scarcer reliable information (Fig 3). It is possible that enzymes used by Mollicutes to integrate lipids in their membranes and assemble glycans into CPS are not very similar to more thoroughly studied proteins in model organisms. Corollary to this hypothesis is the lost ability to synthesize a cell wall, a landmark of Mollicutes evolution (Sirand-Pugnet *et al*, 2007). Therefore, lipid and glycan syntheses are probably performed by currently un-annotated but

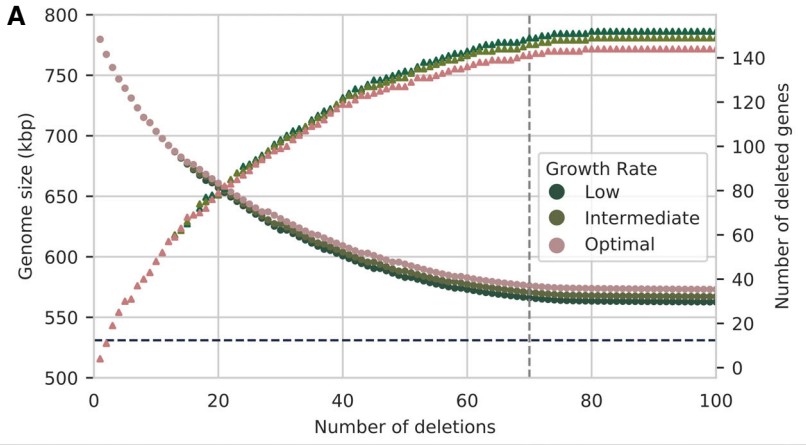

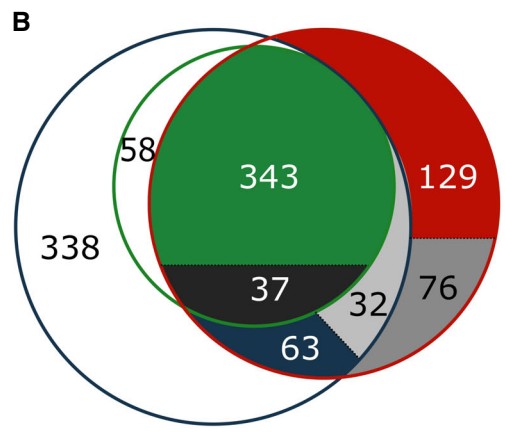

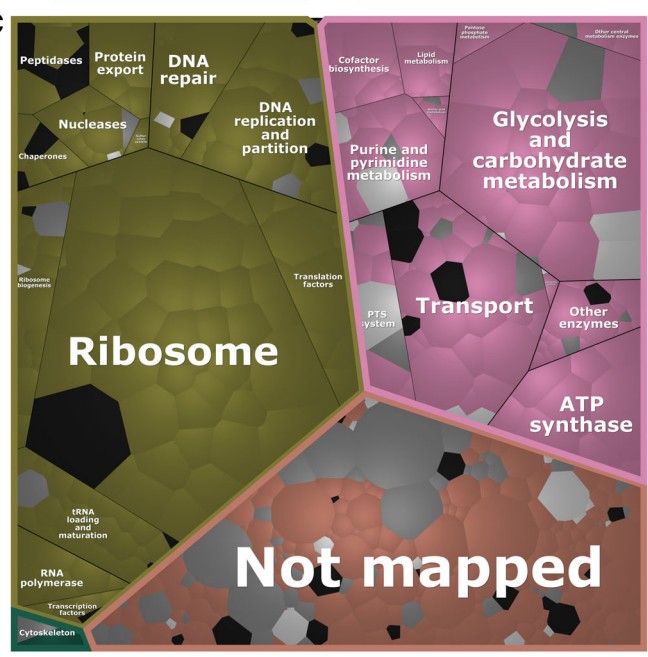

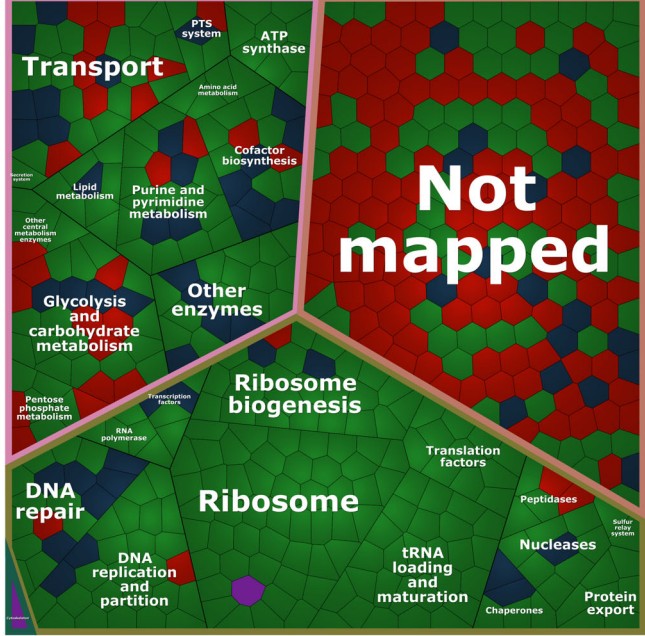

**Figure 7. Model-driven prediction of a minimal genome for *Mesoplasma florum*.**

A Iterative deletions using the MinGenome algorithm. Faded circles represent genome size and bright triangles the number of deleted coding and non-coding genes. Low, intermediate, and optimal growth rate constraints correspond to 60, 90, and 100% of the *M. florum* growth rate measured in CSY (0.44 h⁻¹).

B Venn diagram showing the protein-coding genes shared between *M. florum* (red circle), JCVI-syn3.0 (green circle), and its parent JCVI-syn1.0 (blue circle). The different shades of gray represent the proteins targeted for deletions in the low growth rate constraint presented in (A).

C Voronoi diagram showing the functional distribution of 145 proteins targeted for deletions present in JCVI-syn3.0 (black), *M. florum* only (dark gray), and JCVI-syn1.0 but not JCVI-syn3.0 (light gray). The shapes are sized according to transcriptomic data, and the KEGG categories are represented by bright colors. The single protein associated with environment information processing is represented in black on the top part of the diagram between the genetic information processing and metabolism categories.

D Voronoi diagram showing the functional distribution of the retained proteins in the predicted reduced genome. Shape colors are as in (B), and purple represents proteins shared with JCVI-syn3A but absent in JCVI-syn3.0. Section borders are colored according to KEGG categories depicted in (C).

likely essential enzymes. Additional work will be needed to describe the exact lipid and glycan compositions and the genes involved in their processing.

The need for additional knowledge was also highlighted in our minimal genome prediction (Fig 7 and Dataset EV6). While the majority of the genes targeted for deletion were not mapped to KEGG functional categories, a significant proportion (~ 36%) of the retained proteins were of unknown function (Figs 7C and D, and EV4B). This fraction is very similar to its counterpart in JCVI-syn3.0 where it was initially reported that 149 genes out of the 473 were uncharacterized (~ 32%) (Hutchison et al, 2016).

Among the key features entirely retained in the minimal genome scenario were the cofactor biosynthesis-related proteins (Fig 7C and D), which is consistent with the fact that BOFdat Step 2 enforced the addition of all vitamins and cofactors essential to sustain prokaryotic life (Xavier et al, 2017; Lachance et al, 2019a). Whether or not all these cofactors are effectively necessary to support M. florum's growth could be addressed upon the definition of a completely defined medium. Given that the Vitamin & Cofactors module contained a higher fraction of proteins with a lower confidence level (Fig 3), such a study could provide highly valuable information for the complete understanding of M. florum molecular functions. BOFdat Step 3 was used to find metabolites that contribute to increasing iJL208 gene essentiality prediction when added to the BOF (Fig 5 and Appendix Fig S7). While this approach was not applied to other Mollicutes models, all but one metabolite identified in this step were found in these models' BOF (Table 2). Taken together, these comparisons support the proposed BOF composition for M. florum (Appendix Table S3).

The CSY semi-defined medium allowed the assessment of M. florum growth on various energy sources and identified discrepancies between observations and model predictions (Figs 6 and EV1). Comparing selected 3D protein structures reconstructed with I-TASSER to the PDB using FATCAT 2.0 revealed similarities with proteins of known promiscuity (Fig EV2). While this approach does not constitute a direct validation of enzyme promiscuity, it does provide contextual hypotheses for reducing the search space for eventual biochemical characterizations. Yet our study was not the first to use such ad hoc reconstruction of 3D structures for genome-wide identification of protein functions (Yang & Tsui, 2018; Antczak et al, 2019; Yang et al, 2019). We foresee that faced with the great challenge of identifying numerous molecular functions required for synthetic biology, combining the increasing reliability of structure prediction algorithms (AlQuraishi, 2019; Senior et al, 2019; preprint: Billings et al, 2019) to the predictive power of GEMs is likely to play an important role in organism design in the coming years.

Mesoplasma florum-specific uptake and secretion rates were defined in CSY medium (Fig 4 and Appendix Figs S5 and S6), a measurement performed for only two of the four modeled Mollicutes acknowledged in our study (Bautista et al, 2013; Wodke et al, 2013). While the calculated substrate uptake rate was slightly lower than both values recorded in other Mollicutes, the combined lactate and acetate secretion rate was within the previously measured values (Table 3). In our datasets, the expression of both the LDH (Mfl596) and the PDH (Mfl039-Mfl042) complex-forming genes was observed (Dataset EV5), which led to the hypothesis that both fermentation products would be secreted in M. florum. The initial lactate/acetate secretion rate ratio (8:1) chosen based on the

expression data was exacerbated following sensitivity analysis (~ 15:1). This shift in ratios can be explained by the upper flux limit applied to the NADH oxidase reaction, which utilizes oxygen to recycle NADH cofactor produced when generating acetate, whose addition was necessary to ensure a non-linear relationship between oxygen uptake and growth rate (Fig 5H). This shift also reflected the difference in LDH and PDH expression levels as well as the stoichiometric disparity of the two active protein complexes (Mattevi et al, 1992; Wigley et al, 1992). Our genome reduction scenario conserved the E1 component of the PDH whereas it was absent in JCVI-syn3.0 (Dataset EV6). Hence, the individual and unequivocal quantification of M. florum lactate and acetate secretion rates could reveal the conditions under which this pathway is essential, hereby shedding light on alternate genome reduction paths.

In M. florum, the high proportion of JCVI-syn3.0 orthologs provided an interesting validation for the GEM-driven prediction of a minimal gene set featured in our study (Fig 7). The rational design of minimal genomes using the Mycoplasma genitalium whole-cell model reported minimal genes sets considerably lower than the 473 genes contained in JCVI-syn3.0 (360 and 380) (Rees-Garbutt et al, 2020). While removing individual genes from JCVI-syn3.0 is still possible, combining multiple gene deletions often resulted in greatly reduced growth rates (Breuer et al, 2019; Pelletier et al, 2021). Hence, predictions containing fewer genes than this organism are not likely to be viable. Our reduction scenario contained 90 more genes than JCVI-syn3.0 (563 vs 473), which could be attributed to genuine biological differences between the two organisms or inaccurate predictions given the remaining uncertainties in iJL208 and the resolution in the transposon mutagenesis dataset used by the MinGenome algorithm (Wang & Maranas, 2018).

Comparing our genome reduction scenario to JCVI-syn3.0 revealed the possibility that minimal genomes could use alternate carbohydrates to fuel their cellular needs. We also found that varying the growth rate constraint resulted in a reduced genome more similar to JCVI-syn3.0 (Appendix Fig S10). A growth rate set to 60% of optimal was also notably similar to the growth rate ratio between JCVI-syn3.0 and the more robust JCVI-syn3A (~ 50%) (Breuer et al, 2019). The absence of the E1 complex from JCVI-syn3.0 but its presence in our minimal gene set suggests that varying the constraints imposed on the input GEM could result in different genome reduction scenarios (Dataset EV6). Some proteins from the genetic information processing category differed from JCVI-syn3.0. The impact of these different chaperones, peptidases, ribosome methylases, and ribosome composition (i.e., rpmE) could be assessed by generating a model that includes the expression machinery (ME-model) (Liu et al, 2014; Lloyd et al, 2018).

In conclusion, iJL208 was built on a revised annotation obtained from several computational approaches. Since missing or incomplete information can lead to false or inaccurate predictions, we performed different experiments to validate and increase the overall quality of the model. iJL208 will provide a framework to generate hypotheses, guide future experiments, and reach an exquisite understanding of cellular mechanisms in M. florum. With recent advances enabling complex genome manipulation in M. florum (Matteau et al, 2017; Baby et al, 2018a), iJL208 will also contribute to whole-genome engineering studies in this emerging model organism.

# Materials and Methods

### Bacterial strains and data sources

All experiments described in this study were performed using *M. florum* strain L1 (ATCC 33453). The complete genome sequence of this strain is available in GenBank under RefSeq accession number NC_006055.1. Genome annotations were either based on RefSeq (NC_006055.1), PATRIC (Genome ID: 265311.5), or both depending on specific analysis context and needs. The transposon mutagenesis dataset was taken from Baby *et al* (2018b). *M. florum* biomass composition, gene expression datasets, and lipidomic profile were taken from Matteau and colleagues (Matteau *et al*, 2020). Original transcriptomic and proteomic data are accessible through the Gene Expression Omnibus (GEO) under Series accession number GSE152985 and via the PRIDE partner repository with the dataset identifier PXD019922 and 10.6019/PXD019922, respectively.

### Proteome comparison

The proteome comparison tool from PATRIC (Wattam *et al*, 2017) (https://www.patricbrc.org) was used to identify orthologous proteins between *M. florum* L1 (Genome ID: 265311.5) and the following strains: *Mycoplasma gallisepticum* str. F; Genome ID: 708616.3, *M. pneumoniae* M129; Genome ID: 272634.6, *M. genitalium* G37; Genome ID: 243273.27, and *M. mycoides* JCVI-syn3.0; Genome ID: 2102.8. The parameters to identify orthologous proteins were the following: minimum positives of 0.2, minimum sequence coverage of 0.3, a minimum identity of 0.1, and a maximum *E*-value of 1e5. In the case of pairwise proteome comparisons, both unidirectional and bidirectional best hits are used to define orthologous genes. Gene names were considered similar if they shared the same initial three characters in at least two species.

### Homology modeling

3D protein structures were reconstructed for *M. florum* L1 coding sequences from RefSeq using the I-TASSER Suite 5.1 (Roy *et al*, 2010; Yang *et al*, 2015). To provide relevant homology for functional predictions, a pre-screening step was applied. This step used the Structural Systems Biology software (ssbio) (Mih *et al*, 2018) to compare the sequence of each *M. florum* protein to the PDB of crystallized structures (Berman *et al*, 2000) (www.rcsb.org). HMMER was then used to determine structural domain coverage in the PDB and identified an initial set of 459 proteins with a match to known domains. The following parameters were applied to filter this initial set to determine proteins that were likely to provide reliable structures from homology modeling: *E*-value < 1e-4, domain sequence identity > 10%, and domain sequence similarity > 30%. The transmembrane proteins (20) were also discarded given the limited capability of I-TASSER to produce a relevant model for such proteins (Koehler Leman *et al*, 2015). All in all, a 3D structure was reconstructed for a total of 386 *M. florum* proteins (Dataset EV2). The quality of reconstructed structures is given by the significance of threading template alignments and convergence parameters of the structure assembly simulations (*C*-score) and the similarity between the template and modeled structure (TM-score). The 361 structures

with a "*C*-score" higher than −1.5 and a "TM-score" higher than 0.5 were defined reliable (see Appendix Fig S2).

The FATCAT 2.0 (Li *et al*, 2020) software was used to find similar enzymes to selected structures reconstructed by I-TASSER. Structures were compared against the PDB (90% non-redundant set) using the Database search tool with the flexible FATCAT alignment parameter. Alignments with a *P*-value < 0.05 were considered as significant hits.

### Identification of enzyme commission numbers

Enzyme commission numbers were retrieved from the *M. florum* L1 RefSeq genome annotation (NC_006055.1) using the DETECT v2 (Nursimulu *et al*, 2018) software and from the reliable protein structures reconstructed by I-TASSER using COFACTOR (Zhang *et al*, 2017). Identifications above the default probability threshold from DETECT v2 (90%) were considered as the gold standard. For cross-validation, EC numbers were considered similar if the first three digits were identical in at least two identification methods, i.e., using COFACTOR, DETECT v2, or from the RefSeq and PATRIC annotations.

### Confidence level and final annotation score

A scoring system was established to integrate all information gathered through the computational identification of molecular functions in *M. florum*. For each method, the following score was attributed based on the level of precision that could be attributed to each gene, as presented in Fig 1E: Proteome comparison, Identical = 3, Similar = 2, Different = 1, No gene name or unique = 0; Structural homology, Reliable structure = 3, No reliable structure = 0; EC numbers, Identical EC = 3, Similar EC = 2, Different EC = 1, One method or no EC = 0. The cumulative score attributed to each predicted protein was defined as the final annotation score. Based on this score, a basic (< 3), medium (≥ 3 and < 7), or high (≥ 7) confidence level was attributed.

### Reconstruction of the metabolic network

The draft *M. florum* metabolic model was reconstructed using the SimPheny platform (Genomatica, Inc.) to ensure reaction conformity with a standard database and quality control. The reactions issued from the scaffold generated through the comparative approach were added first. Both RefSeq and PATRIC annotations, along with the identified EC numbers found with DETECT v2 and COFACTOR, were screened manually to identify metabolic candidates. Metabolic reactions associated with these genes were determined based on the information available in public databases (Kanehisa & Goto, 2000; Artimo *et al*, 2012; Kanehisa *et al*, 2016, 2017; Placzek *et al*, 2017). When multiple reactions were possible, preference was given to the terms matching the most detailed genome annotation. Reaction and metabolite names used in the model followed the modeling specific nomenclature of the BiGG database (King *et al*, 2016). The initial SimPheny model was imported in COBRApy (Ebrahim *et al*, 2013) for further manipulations (i.e., addition of species-specific reactions and metabolites, definition of the BOF, flux-balance analysis, etc.). The subsystems from the *E. coli* *i*ML1515 model were assigned to reactions in *i*JL208 when their identifiers had a perfect match with an *i*ML1515

reaction. The subsystems were then grouped together to form the six modules. Reactions that did not match an identifier in the *i*ML1515 model were manually assigned.

## Flux-balance analysis

Flux-balance analysis (FBA) is a mathematical approach to simulate cellular phenotype (Orth *et al*, 2010). The metabolic network is represented as a stoichiometric matrix (*S*) where every row or column represents a unique metabolite or reaction, respectively. The stoichiometry of each metabolite in a reaction is given as a coefficient in the matrix. If we assume a vector of metabolic fluxes *v*, the variation of metabolite concentration over time becomes the following:

$$\frac{dX}{dt} = S \cdot v \tag{1}$$

where *X* is the vector of metabolites in the network. FBA assumes that the metabolic network will reach a steady state. In this case, the concentration of metabolites over time should be in equilibrium where the inputs are equal to the outputs so that:

$$0 = S \cdot v \tag{2}$$

Defining a physiologically meaningful objective (*Z*) allows the formulate an optimization problem on which constraints apply:

$$\begin{aligned} \text{maximize} & Z, \\ 0 &= S \cdot v \\ a_i &< v_i < b_i \end{aligned} \tag{3}$$

where *a* and *b* are the flux bounds on every reaction. This mathematical formulation can be solved using linear programming and allows finding the optimal solution of a given metabolic network at steady state.

## Biomass objective function

The *M. florum* BOF was defined using the BOFdat software (Lachance *et al*, 2019a) and leveraging the previously reported biomass composition of the cell and available omics datasets (transcriptomic, proteomic, and lipidomic) (Matteau *et al*, 2020). The first and second steps of BOFdat were used to determine the precise stoichiometric coefficients of the major cellular macromolecules as well as inorganic ions and coenzymes, respectively. The third step of BOFdat was used to identify metabolites most improving the essentiality prediction accuracy of the model. Revised gene essentiality data previously published for *M. florum* were used in that context (see identification of essential genes section of the Materials and Methods). 50 evolutions were performed for 200 generations each. The Matthews correlation coefficient (MCC) was used to score the biomass compositions. MCC can be calculated using the following confusion matrix:

$$\text{MCC} = \frac{(TP \cdot TN) - (FP \cdot FN)}{\sqrt{(TP+FP)(TP+FN)(TN+FP)(TN+FN)}} \tag{4}$$

The biomass compositions (individuals) with the highest were saved into a "Hall of Fame" for each evolution. The frequency of

apparition of metabolites within the individuals saved in the Hall of Fame was determined, and the ones appearing most frequently were added as part of the BOFdat Step 3. From the nine metabolites identified, seven were considered part of the metabolite pool category (molecular weight fraction [MWF] = 1.2%). The stoichiometric coefficients of every metabolite in this category were re-computed using the BOFdat with this MWF and assuming that each metabolite is represented equally. Two metabolites belonged to the lipids category, so the same procedure was employed but using the lipids MWF (18.3%). Finally, the metabolite representing the *M. florum* CPS was added and its stoichiometric coefficient determined using the carbohydrates MWF (4.1%).

## Development of a semi-defined growth medium

An exponential growth phase *M. florum* preculture grown at 34°C in ATCC 1161 medium (1.75% (w/v) heart infusion broth, 4% (w/v) sucrose, 20% (v/v) HS, 1.35% (w/v) YE, 0.004% (w/v) phenol red, 200 U/ml penicillin G) (Matteau *et al*, 2017) was centrifuged for one min at 21,000 *g* and washed twice with PBS 1X. Washed cells were inoculated at an initial concentration of ~ 1e5 CFU/ml into three different medium bases containing decreasing concentrations of HS and YE (from 20% HS/1.35% YE to 0.01% HS/ 0.0006% YE) and either 1.75% (w/v) heart infusion broth (ATCC 1161 base), PBS 1X (PBS base), or CMRL-1066 chemically defined medium (C5900-02A, US Biological; CMRL-1066 base), all supplemented with 0.004% (w/v) phenol red and 200 U/ml penicillin G. Medium bases were adjusted to a pH of ~ 7.5 and transferred into a 96-well microplate for growth assays. Half of wells were also supplemented with sucrose at a final concentration of 4% (w/v) for the ATCC 1161 base and 1% (w/v) for PBS and CMRL-1066 medium bases. The inoculated microplate was incubated with shaking at 34°C in a Multiskan GO microplate reader (Thermo Scientific), and the optical density at 560 nm ($OD_{560\,nm}$) was measured every 10 min for 24 h. Changes in the absorbance of phenol red at 560 nm caused by the metabolic activity of *M. florum* (medium acidification) were previously shown to correlate with the number of CFU/ml (Matteau *et al*, 2015, 2020). Color fold change was then evaluated by comparing the minimal $OD_{560\,nm}$ value observed over the entire incubation period to a non-inoculated control of identical medium composition. The color fold change observed for sucrose-containing wells was then compared to wells not supplemented with sucrose, resulting in a normalized growth index for each tested medium base. The medium composition showing the highest normalized growth index for the lowest concentrations of HS and YE (CMRL-1066 base supplemented with 0.313% HS and 0.02% YE, Appendix Fig S3), referred to as CSY, was selected for the evaluation of growth sustaining carbohydrates. All conditions were tested in technical triplicate.

## Experimental evaluation of *M. florum* growth on different carbohydrates

A 96-well microplate was filled with CSY supplemented with either one of the following carbohydrates, at a final concentration of 1% (w/v): sucrose, trehalose, fructose, glucose, maltose, mannose, glycerol, sorbitol, lactose, galactose, ribose, arabinose, *N*-acetylglucosamine, and glycerol-3-phosphate. A no-sugar control was also performed. Half of wells were inoculated at an initial

concentration of ~ 1e5 CFU/ml with an *M. florum* preculture prepared and washed as described in the previous section. The microplate was incubated at 34°C without shaking for 24 h. The $OD_{560\ nm}$ at 24 h was measured using a Multiskan GO microplate reader (Thermo Scientific) and compared between inoculated and non-inoculated conditions, resulting in a growth index for each carbohydrate tested. All conditions were tested in technical triplicate. For comparison with the model's predictions, the growth index measured for each carbohydrate was finally normalized to growth on sucrose.

### *In silico* prediction of carbohydrate utilization

The formulated model with defined biomass composition and constraints was used for the prediction of carbohydrate utilization. An exchange reaction, the model equivalent of the medium composition was added for each carbohydrate tested experimentally. When tested, a lower bound of −10 was applied to the exchange reaction and the model was optimized for biomass production. The predicted growth rates for each carbohydrate were saved and normalized over sucrose to facilitate comparison with experimental results.

### Measurement of *M. florum* doubling time and growth rate calculation

The doubling time of isolated transposon insertion mutants as well as *M. florum* growing in CMRL-1066 base medium with variable HS and YE concentrations was measured using colorimetric assays as described previously (Matteau *et al*, 2020). Cultures were performed in technical duplicate and incubated at 34°C with shaking. Growth data of insertion mutants are available in Dataset EV5. For *M. florum* growing in CSY medium (0.313% HS and 0.02% YE) with variable initial concentration of sucrose, the doubling time was measured according to CFU counts of time-course experiments. Briefly, a simple exponential growth model was fit to the mean CFU counts measured over time for each initial sucrose concentration (equation (5)):

$$A_t = A_0 e^{rt} \tag{5}$$

where $A_0$ is the initial number of bacteria and $r$ is the growth rate. In simple exponential growth, the relation between growth rate ($r$) and doubling time ($d$) is given by:

$$d = \frac{\ln(2)}{r} \tag{6}$$

### Quantification of sucrose uptake rate and fermentation product secretion rate

An exponential growth phase *M. florum* preculture grown at 34°C in ATCC 1161 medium was centrifuged for one min at 21,000 *g* and washed twice with PBS 1X. Washed cells were inoculated at an initial concentration of ~ 1e5 CFU/ml into different CSY media containing variable concentrations of sucrose. Inoculated media were adjusted to a pH of ~ 7.5 and transferred into a 96-well microplate for growth experiments. Cultures were incubated with shaking at 34°C in a Multiskan GO microplate reader (Thermo Scientific),

and growth was followed by measuring CFU counts every ~ 90–120 min until late exponential phase. CFU were evaluated by spotting serial dilutions of the cultures on ATCC 1161 solid medium and counting colonies after an incubation of 24–48 h at 34°C. For modeling purposes, CFU/ml were converted to gDW/l (biomass) according to the previously determined *M. florum* dry weight (Matteau *et al*, 2020). In addition to CFU/ml measurements, sucrose and fermentation products (lactate and acetate) were also quantified throughout the entire experiments by HPLC. For this task, cultures were filtered through 0.2-μM PES filters and frozen at −80°C until quantification. HPLC analysis was performed by the Laboratoire des Technologies de la Biomasse at the Université de Sherbrooke. A Dionex CarboPac SA10–4 μM column was used for sucrose quantification, while a Dionex IonPac AS11-HC-4μm IC column was used for lactate and acetate quantification. The injection volume was set to 5 μl, and both the electrochemical detector and columns were operated at a temperature of 30°C. Mobile phase was composed of aqueous KOH solution, and the elution gradient mode was set as follows: sucrose, 1 mM for 12 min, 10 mM for 5 min, and 1 mM for 10 min; acetate and lactate, 1 mM for 5 min, 15 mM for 9 min, and 30 mM for 11 min. The flow rate was maintained at 1.25 ml/min for sucrose and 1.5 ml/min for lactate and acetate. For sucrose quantification, the stability of the signal was ensured by a 200 mM KOH post-injection using a Dionex GP 50 gradient pump set to 0.25 ml/min. Quantifications were performed by external calibration using 99.95% sucrose (Acros), 98% anhydrous L-lactic acid (Alfa Aesar), and 99.7% acetic acid (Fluka). Since lactate and acetate peaks were indiscernible, corresponding peak areas were combined, resulting in a combined lactate/acetate estimate. Following quantification, substrate-specific (sucrose) uptake rate and fermentation product-specific (lactate/acetate) secretion rates were calculated according to the following equation:

$$qS = \frac{\Delta S \cdot r}{X_{t2}} \tag{7}$$

where $qS$ is the substrate (or product)-specific rate, $\Delta S$ is the variation of substrate concentration over time and $X_{t2}$ is the biomass concentration at the end of the time interval (Sauer *et al*, 1999). To calculate $qS$, simple exponential fits were applied to the mean of biomass, sucrose, and lactate/acetate concentration data points (Appendix Fig S5). A linear regression in exponential phase (14–16 h) was then applied to these fits (Appendix Fig S6), and $\Delta S$ and $X_{t2}$ were calculated for each 1-h time interval using the parameters of these regressions. Growth rates ($r$) were obtained from the exponential fits applied to biomass data of each condition (equation (5)). The maximum substrate and product-specific rates are expected to reach a plateau as the initial concentration of substrate increases. We estimated that this plateau would be reached at the two highest initial sucrose concentrations tested in this study (0.05 and 0.1%). For modeling purposes, substrate- and product-specific rates were therefore determined by computing the average of the possible rates obtained for these two initial sucrose concentrations.

### Sensitivity analysis

The model sensitivity to maintenance costs as well as uptake and secretion rates was assessed by setting initial parameters and further

varying each of them individually. The initial parameters used were as follows: GAM: −5 mmol/gDW/h, NGAM: 3 mmol/gDW/h, sucrose uptake rate: −5.26 mmol/gDW/h, lactate secretion rate: −7.65 mmol/gDW/h, acetate secretion rate: −0.96 mmol/gDW/h, and oxygen uptake rate: −10 mmol/gDW/h. The initial sucrose uptake rate used is the rate defined experimentally in CSY medium. The sum of the initial lactate and acetate secretion rates is equal to the combined secretion rate measured experimentally at −8.69 mmol/gDW/h. The choice was made to favor lactate over acetate secretion since its path to secretion in the metabolic network is simpler. The initial oxygen uptake rate represents half the rate reported for *E. coli* growing in minimal medium batch cultures (Andersen & von Meyenburg, 1980).

Maintenance costs represent the amount of energy necessary to support the cell aside from the production of biomass components from the metabolic network. While GAM is the ATP hydrolysis reaction within the BOF, the NGAM is represented by the ATP maintenance reaction which also consumes ATP but is independent of biomass production. These rates were defined first as they are most impactful for growth rate prediction (Lachance *et al*, 2019a). Using the initial parameters, the GAM was determined by varying the ATP maintenance in the BOF from 0 to 50 mmol/gDW/h and predicting the growth rate with standard FBA optimization for each GAM value. The theoretical GAM value was identified by matching the experimental growth rate in CSY (0.44 h$^{-1}$). Similarly, the NGAM was identified by varying its value from 0 to 30 mmol/gDW/h, fixing the theoretical GAM and keeping the initial uptake and secretion rates. This allowed the identification of the NGAM value at which the predicted growth rate fits its experimental counterpart.

The model sensitivity to uptake and secretion rates was then evaluated using the fixed theoretical maintenance costs. Using the initial oxygen uptake as well as lactate and acetate secretion rates, the sucrose uptake rate was varied between 3 and 15 mmol/gDW/h, which encompasses the experimentally measured uptake rate of 5.26 mmol/gDW/h (absolute value). Sensitivity to lactate and acetate production was evaluated on specific ranges (0–40, 0–15 mmol/gDW/h, respectively) with the determined sucrose uptake rate and initial oxygen uptake rate. The lactate secretion rate was determined as the minimum rate matching the experimental growth rate in CSY (0.44 h$^{-1}$). This value was found at 8.16 mmol/gDW/h. The acetate secretion rate was defined as the difference between the average experimentally determined value of 8.69 mmol/gDW/h for both products and the theoretical lactate secretion rate, corresponding to 0.53 mmol/gDW/h. Finally, the impact of oxygen uptake rate was assessed using all constraints and rates previously determined. The oxygen uptake rate varied between 0 and 30 mmol/gDW/h. The final uptake rate (4.81 mmol/gDW/h) was chosen to match the experimental growth rate value in CSY (0.44 h$^{-1}$).

## Identification of expressed genes

Transcriptomic and proteomic expression datasets were available for *M. florum* (Matteau *et al*, 2020). To determine the set of expressed genes, the reported number of protein molecules per cell and fragments per kilobase per million of mapped reads (FPKM) associated with each gene was compared with each other. An expression threshold spanning the entire range of measured values was iteratively applied to each dataset resulting in a list of expressed

and unexpressed genes. The resulting binary vectors were compared using the MCC by setting the transcriptomic data as a reference and generating a distinct MCC value per pair of thresholds. A correlation score ($S_{i,j}$) was obtained by multiplying the MCC score for each threshold pair ($M_{i,j}$) by the number of genes expressed at these same thresholds ($X_{i,j}$):

$$S_{i,j} = X_{i,j} \cdot M_{i,j} \tag{8}$$

This score accounts for the correlation between each dataset while maximizing the number of expressed genes. The thresholds providing the optimal score were used for this study and were found at 23 proteins per cell (proteomics) and 168 FPKM (transcriptomics).

## Identification of essential genes

Previously published experimental essentiality data (Baby *et al*, 2018b) generated by transposon mutagenesis were re-analyzed in this study. The doubling time measured for individual mutants (Dataset EV5) was used along with the relative position of the insertion site within the interrupted gene to re-evaluate gene essentiality (Fig EV3). The growth data was filtered to include only mutants for which the standard deviation of doubling time between replicates was within 30% of the average doubling time measured. Mutants for which a reliable doubling time could be obtained were defined as non-viable if their measured doubling time exceeded the sum of the median and median absolute deviation. For insertions not impairing the growth of *M. florum*, interrupted genes were considered essential only if the transposons were strictly restricted to the terminal region of genes, defined as the last 20% of the gene length. Both Hutchison and colleagues (Hutchison *et al*, 2016) and Breuer and colleagues (Breuer *et al*, 2019) used the location of transposon insertion to nuance their essentiality observations.

## Prediction of metabolic flux state

The flux state through the metabolic network was obtained by optimizing the production of biomass using parsimonious flux-balance analysis (pFBA), a version of FBA that allows the generation of a unique flux state prediction through minimization of enzyme usage (Lewis *et al*, 2010). This method is best suited for the comparison of predicted fluxes to gene expression (Machado & Herrgård, 2014). A reaction flux was defined as active when the predicted value exceeded the numerical error (1e8) and the flux was attributed to every gene that could catalyze the reaction via the gene-reaction rule. The objective was set to the BOF from BOFdat Step 3 and the *in silico* medium (Appendix Table S2) set with sucrose as the main energy source.

## Model-driven prediction of a minimal gene set and identification of functional features

The MinGenome algorithm (Wang & Maranas, 2018) was used to sequentially identify the longest possible deletions in the *M. florum* genome. The transcription units (relationship between gene and promoter locations) were obtained from the integrative characterization of *M. florum* (Matteau *et al*, 2020). The *i*JL208 model along

with the experimentally determined essential genes revised in this study was also used as input. The algorithm extracts constraints from these inputs and writes a bi-level linear program where the lower level optimizes the production of biomass and the higher level maximizes the DNA length (bp) of the deletion to be performed. To identify the minimal gene set for *M. florum*, the optimization was performed iteratively 100 times. Deleted genes and promoters were encompassed between a deletion start and end site.

The *M. florum* proteome was compared to that of JCVI-syn1.0 (FASTA file reconstructed from DataSetS1 available in Hutchison *et al* (2016)) and JCVI-syn3.0 (Genome ID: 2102.8) using the PATRIC proteome comparison. Since multiple comparisons were executed, only bidirectional best hits were used to define homologous genes. Mapping of *M. florum* genes to KEGG functional categories (Kanehisa *et al*, 2004) was performed as described previously (Matteau *et al*, 2020), where the automated attributions were manually curated to fit the context of *M. florum*. The composition of the cell was depicted using the proteomap software (Liebermeister *et al*, 2014), and the protein abundance was obtained from available transcriptomic data (Matteau *et al*, 2020).

## Data availability

The final version of *i*JL208 was processed through the Memote software (Lieven *et al*, 2020) to ensure compliance with the current standards for metabolic modeling. This report, the final *i*JL208 along with all code necessary to generate the results presented in this study is available on GitHub (https://github.com/jclachance/iJL208). The final *i*JL208 model is also available in JSON format as Code EV2. An interactive map of the entire reconstructed *M. florum* metabolic network was built using Escher (King *et al*, 2015) and is provided as Code EV1. The central metabolism map of the *E. coli* *i*JO1366 model was used as an initial template on which the *i*JL208 model was mapped. The map was manually expanded using the reactions available in the model.

**Expanded View** for this article is available online.

### Acknowledgements
Funding for this research was provided by the Natural Sciences and Engineering Research Council of Canada: 2020-06151, by the Fonds de recherche du Québec—Nature et technologies: 2018-PR-206064, and by the Novo Nordisk Foundation: NNF10CC1016517. The authors would like to acknowledge and thank for their indirect but substantial contribution to this work, members of the Systems Biology Research Group that were not listed as authors; specifically Dr. Jared Broddrick, Dr. Erol Kavvas, Dr. Yara Seif, and Charles J. Norsigian for the multiple computational modeling-related discussions. Edward Catoiu for additional support with 3D modeling of protein structures; and Marc Abrams for the revision of written English. A special thanks to Pr. Laurence Yang and Dr. Anand V. Sastry for conceptualization of the study.

### Author contributions
J-CL conceptualized the study, curated the data, involved in formal analysis and in funding acquisition, investigated the study, contributed to methodology and visualization of the study, developed software, validated the study, wrote the original draft, reviewed and edited the manuscript. DM conceptualized the study, curated the data, investigated the study, contributed to methodology and visualization of the study, validated the study, reviewed and edited the manuscript. JB investigated and validated the study. NM involved in formal analysis, curated the data, investigated the study, and contributed to methodology. CJL conceptualized and validated the study. ZAK supervised and validated the study. TFK investigated and curated the data. AMF provided resources and involved in funding acquisition. JMM conceptualized, supervised, and provided resources. BOP conceptualized and supervised the study, provided resources, and involved in funding acquisition. P-ÉJ conceptualized and supervised the study, provided resources, involved in funding acquisition, and reviewed and edited the manuscript. SR supervised project administration, conceptualized and supervised the study, provided resources, involved in funding acquisition, and reviewed and edited the manuscript.

### Conflict of interest
The authors declare that they have no conflict of interest.

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
