## [Review Process File · Molecular Systems Biology]

Genome-scale metabolic modeling reveals key features of a minimal gene set

Jean-Christophe Lachance, Dominick Matteau, Joëlle Brodeur, Colton Lloyd, Nathan Mih, Zachary King, Tom Knight, Adam Feist, Jonathan Monk, Bernhard Palsson, Pierre-Étienne Jacques, and Sébastien Rodrigue

DOI: [10.15252/msb.202010099](https://doi.org/10.15252/msb.202010099)

Corresponding author(s): Sébastien Rodrigue (Sebastien.Rodrigue@USherbrooke.ca)

Review Timeline:

Submission Date:	16th Nov 20
Editorial Decision:	14th Jan 21
Revision Received:	11th May 21
Editorial Decision:	9th Jun 21
Revision Received:	18th Jun 21
Accepted:	22nd Jun 21

Editor: Jingyi Hou

Transaction Report:

Thank you for submitting your work to Molecular Systems Biology. We have now heard back from two of the three reviewers who agreed to evaluate your study. Unfortunately, after a series of reminders we did not manage to obtain a report from Reviewer #1. In the interest of time, and since the recommendations of Reviewers #2 and #3 are quite similar, we have decided to proceed with these two reports. As you will see below, the reviewers acknowledge that the presented findings and data are potentially useful for the field. They raise however a series of -mostly minor- concerns, which we would ask you to address in a major revision.

I think that the reviewers' recommendations are rather clear and there is therefore no need to reiterate the comments listed below. All issues raised by the reviewers need to be satisfactorily addressed. Please feel free to contact me in case you would like to discuss in further detail any of the issues raised.

On a more editorial level, please do the following:

REFeree REPORTS

Reviewer #2:

The manuscript "Genome-scale metabolic modeling reveals key features of a minimal gene set" is another step in the continuing brilliant work investigating basic principles of cellular life by the *Mesoplasma florum* team led by Sébastien Rodrigue. This paper will be the standard by which

genome scale computational modeling papers are judged going forward. The amount of data presented here is huge and as a result, much of the information presented is offered with little explanation. The project described might better be presented as a book rather than a stand alone article.

I do take issue with the authors' suggestion that the gene currently annotated as a 2-deoxyribose-5-phosphate aldolase (gene Mfl121 aka deoC) actually encodes a transaldolase. As noted by the authors, no mycoplasma has an annotated transaldolase gene; however, the authors' computational model of *M. florum* predicts that the cell encodes one. The same is true in Breuer et al.'s model of JCVI-syn3A. I redid the I-TASSER reconstruction of Mfl121 protein against the PDB as the authors say they did. They reported Mfl121 was similar to the *Thermatoga maritima* transaldolase, that was not the closest hit. Calling Mfl121 a transaldolase instead of or in addition to it being a 2-deoxyribose-5-phosphate aldolase seems like a major stretch. Furthermore, in these authors' 2018 transposon bombardment study, Mfl121 was not essential. Based on their whole cell computational model, I would think transaldolase would either be essential, or its disruption would have greatly slowed *M. florum* growth. Perhaps more likely is that the model is incorrect and instead of a transaldolase, mycoplasmas employ sedoheptulose-1,7-bisphosphatase drive the pentose phosphate pathway. I urge the authors to rethink their hypotheses about transaldolase.

In my view, this paper can be published with only minor modifications. The writing is solid, and points are clear. Below I list items I would like the authors to consider addressing.

Minor issues that the authors should consider addressing

Page 3 Lines 52-54 The sentence "The capability to physically write entire chromosomes from synthetic DNA is an outset for synthetic genomics, but the ability to predict whether or not this assembly will produce a viable cell remains a substantial challenge." is hard to read. Perhaps you should change this to something more like "The capability to physically write entire chromosomes from synthetic DNA is now an achieved ambition of synthetic genomics, but the ability to predict whether or not this assembly will produce a viable cell remains a substantial challenge."

Page 3 Line 60 JCVI-syn3.0A is incorrect. This should be JCVI-syn3A

Page 3 Line 72 Markus Covert's 2012 model of *M. genitalium* should be referenced (Karr JR, Sanghvi JC, Macklin DN, Gutschow MV, Jacobs JM, Bolival B, Jr., et al. A whole-cell computational model predicts phenotype from genotype. *Cell*. 2012;150(2):389-401.) and Masaru Tomita E-cell paper from 2001 should be listed here (Tomita M. Whole-cell simulation: a grand challenge of the 21st century. *Trends in biotechnology*. 2001;19(6):205-10.)

Page 7 Lines 144-146 The sentence that begins "Interestingly, about 70%..." should instead perhaps begin "Not unexpectedly..." because *M. florum* is in the mycoides clade of mycoplasmas as is *M. mycoides*, the progenitor of JCVI-syn3A. The other mycoplasma GEMS listed are all from the pneumoniae clade of mycoplasma phylogeny, which is well removed from the mycoides clade.

Page 10 Lines 212-213 With regards to the sentence "Among these 10, a 212 single gene was wrongfully identified as essential compared to transposon mutagenesis data." Knowing what gene was misidentified would be interesting to know.

Page 12 Lines 279-282 The sentence "In parallel, the original transposon mutagenesis data were reanalyzed, resulting in the re-assignment of 79 coding genes previously considered as non-essential, for a total of 332 *M. florum* genes determined as essential (Supplementary figure 11,

Supplementary file 9, and Material and methods)." Needs more explanation. If a gene was disrupted by a transposon, then it cannot be essential.

Page 21 Line 452-455 Mentioned in their 2019 eLife paper "Essential metabolism for a minimal cell" and their 2020 BioRxiv paper "Genetic requirements for cell division in a genomically minimal cell" the JCVI-syn3A team reports a number of genes in that organism that can be removed individually although deletion of multiple seemingly unrelated genes sometimes greatly slowed growth rates. That said, I suggest rewording the sentence in this manuscript "Given that JCVI-syn3.0 was pleomorphic, had a doubling time three times greater than its parent JCVI-syn1.0, and that 19 genes had to be re-inserted to produce the more robust JCVI-syn3.0A 8, predictions containing fewer genes than this organism are not likely to be viable."

Reviewer #3:

In this paper, the authors present iJL208, the first genome-scale model (GEM) of metabolism for *Mesoplasma florum*, a fast-growing near-minimal organism. The model accounts for ~30% of the protein-coding genes. The authors have used computational approaches that rely on both sequence and structural homology to investigate and review the gene annotations. The three computational methods, proteome comparison, structural homology, and enzyme commission (EC) number identification, were combined to assign a final annotation score for each *M. florum* protein. A novel semi-defined growth medium was developed to identify the primary energy sources metabolized by *M. florum*. The model constraints for substrate uptake and product secretion rates were determined in the novel medium. Genome-wide expressions and essentiality datasets and the growth data on different carbohydrates were used to validate and refine iJL208. The flux states and gene essentiality predictions were made to validate the model against genome-wide expression and essentiality datasets. Finally, the authors used the phylogenetic proximity of *M. florum* with the minimal cell *Mycoplasma mycoides* JCVI-syn3.0 to assess the predictive power of GEMs for designing minimal genomes. They used an experimentally validated iJL208 model to formulate a minimal genome prediction. It can account for both transcription unit architecture and genome-wide essentiality.

Comments

Major

- Line 125, It was mentioned that the pseudo-reactions (95) were excluded while calculating the fraction of reactions in the model that are transport reactions. The denominator of the fraction of transport reactions is mentioned 278, which is not consistent if we consider a total of 370 reactions (excluding pseudo-reaction will give 275 reactions). Can the author provide the details of the 84-transport reaction and 278 reactions, if any excluded reactions are considered exceptions?
- Page 6: Line 121: Fig 2, Can the author provide the details of pseudo-reactions in the figure? What is the consideration of such reactions in GEM models? Line 121 suggests 236/370 gene associated reactions, but Fig. 2 shows a count of 275 reactions. The discrepancy is due to the consideration of 39 CO₂ transport reactions in the module reactions in fig. 2. The author must include the details of transport reactions in the figure to avoid such issues. Line 128-129 suggests that the iJL208 reactions were grouped into six different modules (275 reactions). No information about pseudo-reactions is mentioned.
- Page 7: Line 135-142: Fig. 3A, Can the author provide the proportion of orphan reactions for six modules. Fig. 3A does not suggest glycan in the list of the highest percentage of orphan relative to the module's total reaction. A quantitative description in the figure will be required to corroborate

text with the figure.

- Line 144-149: table 1 suggests that the total number of genes in the model is 680, but Line 146 mentions 685 genes. The author must correct the number of genes in the model, either in table 1 or the paragraph, to make the data consistent.
- Fig. 7B, The Venn diagram showed does not justify the numbers mentioned in it. Brief labeling alongside the numbers in the Venn diagram can give more insight into the data. For example, the number of deletions shown in the Venn diagram is $(37+32+76 = 145)$, which does not match with the number (152) mentioned in Line 310. And a similar case with the conserved proteins. The author must correct the data shown in the Fig. 7B. There is a missing color label for environment information processing (EIP) in Fig. 7C mentioned only in the figure caption. The author must correct the labeling issues.

Minor

- Fig. 1F, Line 104, the number of proteins mentioned in the result section for different confidence intervals does not look the same as shown in the plot. Can the author provide the count for each final annotation score along with the histogram?
- Line 382: The author needs to describe the PTS system before using the abbreviation.
- Table 3, * and ** shows extrapolation from other species and mollicutes models, but no method was demonstrated.
- Line 518: Homology Modeling: C-score and Tm-scores were not described. The author must describe those scores and what it signifies.
- Line 593, 751: No description of MCC was shown in the method section. The author must provide that.
- Line 619: No reference for previous work
- Line 659: the line specifies growth rate as k , whereas eq. 4 and 5 consider r as growth rate. No use of k is mentioned anywhere else in the article.
- Line 683: what column is XXX must be described.
- Line 701-702: What is the other sucrose concentration that was considered? Can the author provide the variation in the maximum substrate and product-specific rates for different sucrose concentrations? Also, give the reasoning behind considering only the two highest initial sucrose concentration.
- Page 34, Eqn. 7, The author must describe the parameters used in the equation
- The authors must fix the formatting of the equations.

Reviewer #2:

The manuscript "Genome-scale metabolic modeling reveals key features of a minimal gene set" is another step in the continuing brilliant work investigating basic principles of cellular life by the *Mesoplasma florum* team led by Sébastien Rodrigue. This paper will be the standard by which genome scale computational modeling papers are judged going forward. The amount of data presented here is huge and as a result, much of the information presented is offered with little explanation. The project described might better be presented as a book rather than a stand alone article.

First of all, we would like to thank Reviewer #2 for his kind comments about our work and for the time dedicated to the revision of our manuscript. Please find below our point-by-point response in blue, with sentences from the revised manuscript in red.

Major issues that the authors should consider addressing

I do take issue with the authors' suggestion that the gene currently annotated as a 2-deoxyribose-5-phosphate aldolase (gene Mfl121 aka deoC) actually encodes a transaldolase. As noted by the authors, no mycoplasma has an annotated transaldolase gene; however, the authors' computational model of *M. florum* predicts that the cell encodes one. The same is true in Breuer *et al.*'s model of JCVI-syn3A. I redid the I-TASSER reconstruction of Mfl121 protein against the PDB as the authors say they did. They reported Mfl121 was similar to the *Thermotoga maritima* transaldolase, that was not the closest hit. Calling Mfl121 a transaldolase instead of or in addition to it being a 2-deoxyribose-5-phosphate aldolase seems like a major stretch. Furthermore, in these authors' 2018 transposon bombardment study, Mfl121 was not essential. Based on their whole cell computational model, I would think transaldolase would either be essential, or its disruption would have greatly slowed *M. florum* growth. Perhaps more likely is that the model is incorrect and instead of a transaldolase, mycoplasmas employ sedoheptulose-1,7-bisphosphatase to drive the pentose phosphate pathway. I urge the authors to rethink their hypotheses about transaldolase.

The suggestion that the *Mesoplasma florum* pentose phosphate pathway (PPP) contains a transaldolase was evaluated for a long time before assigning this function to the *mfl121* gene product. We considered that it was important to raise this question in our model/publication since the absence of this enzyme is a particular feature of Mollicutes that needs greater attention. We divided the reviewer's comment into 3 main concerns:

1- *Thermotoga maritima*'s transaldolase is not the closest structural hit for Mfl121

Mfl121 shows homology to many homologs as seen in Figure EV2D. Since Mfl121 is annotated as a 2-deoxyribose-5-phosphate aldolase, it is not surprising that structural comparison-based approaches also yield closer matches with proteins bearing this annotation. However, a problem of current's annotation pipelines is that they generally yield a single annotation although it is possible that enzymes also carry other activities. We hypothesize that this could be the case here, and that Mfl121 could have the 2-

deoxyribose-5-phosphate aldolase and transaldolase activities, which would explain the similarity ($p = 5.96e-10$) with *Thermotoga maritima*'s transaldolase.

2- Non-Essentiality of *mfl121*

As noted by the reviewer, *mfl121* is not an essential gene but this could be explained by two main reasons:

- A) Growth condition context:** as shown in Figure 6, none of the PPP genes are essential but are expressed. This suggests that the PPP is an important pathway for Mollicutes but that, depending on the growth conditions, alternative strategies are possible. A growth medium that is sufficiently rich in fully accessible nucleotides (including the ribose backbone) may alleviate the requirement for the PPP. We aim at validating such hypotheses in a completely defined medium with different nucleotide species.
- B) Existence of a paralog:** The reviewer's comment prompted us to re-investigate the *M. florum* gene set in the search for other genes that could act as a transaldolase. The *mfl639* gene is also identified as a 2-deoxyribose-5-phosphate aldolase. We thus compared the predicted structures of Mfl121 and Mfl639 with the overlaid now presented in Figure EV2F. Both proteins show an almost perfect similitude (p -value = 0.00), suggesting that both genes could execute the same tasks. This functional redundancy could explain the non-essentiality of *mfl121* and *mfl639* independently although their simultaneous inactivation could be lethal. Given its potential role as a transaldolase, Mfl639 was added to the gene reaction rule of TALA in the model (see File EV4) and in Figure 6E. Mfl639 was also screened against the entire PDB-90 database and the results are shown in Figure EV2E and available in File EV7. As for Mfl121, the structure of *Thermotoga maritima*'s transaldolase (1vpxA) figured among the top hits ($p = 2.51e-09$) and had the highest similarity of all transaldolases identified.

3- Instead of a promiscuous reaction, a sedoheptulose-bisphosphatase drives the pentose phosphate pathway in Mollicutes.

Through our revision of the *M. florum* annotation, and given the absence of a predicted transaldolase, we considered different paths that could be taken for sedoheptulose, which has been detected in other Mollicutes by metabolomic experiments (Vanyushkina AA, Fisunov GY, Gorbachev AY, Kamashev DE, Govorun VM. 2014 Metabolomic Analysis of Three Mollicute Species. PLoS ONE 9(3): e89312. <https://doi.org/10.1371/journal.pone.0089312>). Unfortunately, the absence of a completely defined medium currently hampers our ability to address this question. In addition, we have found no evidence of a sedoheptulose-bisphosphatase in the *M. florum* annotation or in the predicted protein structures. The structural evidence provided in this study with Mfl121 and Mfl639 are, to the best of our knowledge, the most likely mechanistic explanation that could be provided. We would be happy to further investigate the possibility that a sedoheptulose-1,7-bisphosphatase is encoded by *M. florum* if the reviewer has additional insights that could help us identify this enzyme in Mollicutes.

In sum, we wish to thank the reviewer for this valuable comment that made us reconsider our current hypothesis. In addition of including Mfl639 homology results in File EV7, Figure 6, and Figure EV2 (and modifying corresponding figure legends), we have modified the results section of the manuscript to introduce both Mfl121 and Mfl639 as potential transaldolases (lines 287-294). We now also clearly indicate that this is a working hypothesis that will require experimental validations:

"Here, the I-TASSER reconstructed structure of two 2-deoxyribose-5-phosphate aldolases (Mfl121 and Mfl639) were queried against the PDB to find potential matches with transaldolase structures (Figure EV2D-F and File EV7). Of the 12 transaldolases identified, the structure from *Thermotoga maritima* had the most significant match and highest similarity (Figure 6E and Appendix Table S4). While this structural similarity points to a potential transaldolase reaction, experimental validation will be required to confirm its presence. Meanwhile, the TALA reaction was assigned to Mfl121 or Mfl639 in *iJL208*, thereby allowing flux through the PPP, which is consistent with expression data."

Mfl639 was also added in Table S4 of the Appendix and the Appendix Supplementary Text was modified accordingly (modifications in red).

In my view, this paper can be published with only minor modifications. The writing is solid, and points are clear. Below I list items I would like the authors to consider addressing.

We thank the reviewer for his/her constructive comments that improved our manuscript.

Minor issues that the authors should consider addressing

Page 3 Lines 52-54 The sentence "The capability to physically write entire chromosomes from synthetic DNA is an outset for synthetic genomics, but the ability to predict whether or not this assembly will produce a viable cell remains a substantial challenge." is hard to read. Perhaps you should change this to something more like "The capability to physically write entire chromosomes from synthetic DNA is now an achieved ambition of synthetic genomics, but the ability to predict whether or not this assembly will produce a viable cell remains a substantial challenge."

The readability was improved by changing the sentence as suggested by the reviewer (lines 48-50).

Page 3 Line 60 JCVI-syn3.0A is incorrect. This should be JCVI-syn3A

All instances of JCVI-syn3.0A were changed to JCVI-syn3A in the Main text, Tables, Figure legends, and in the Appendix Supplementary Text.

Page 3 Line 72 Markus Covert's 2012 model of *M. genitalium* should be referenced (Karr JR, Sanghvi JC, Macklin DN, Gutschow MV, Jacobs JM, Bolival B, Jr., et al. A whole-cell computational model predicts

phenotype from genotype. Cell. 2012;150(2):389-401.) and Masaru Tomita E-cell paper from 2001 should be listed here (Tomita M. Whole-cell simulation: a grand challenge of the 21st century. Trends in biotechnology. 2001;19(6):205-10.)

These references were added as suggested (lines 70-71).

Page 7 Lines 144-146 The sentence that begins "Interestingly, about 70%..." should instead perhaps begin "Not unexpectedly..." because *M. florum* is in the mycoides clade of mycoplasmas as is *M. mycoides*, the progenitor of JCVI-syn3A. The other mycoplasma GEMS listed are all from the pneumoniae clade of mycoplasma phylogeny, which is well removed from the mycoides clade.

"Interestingly" was changed to "As expected" at line 139 (to avoid double negation).

Page 10 Lines 212-213 With regards to the sentence "Among these 10, a single gene was wrongfully identified as essential compared to transposon mutagenesis data." Knowing what gene was misidentified would be interesting to know.

While the referenced gene was shown in Figure 5B, we understand that this information can easily go unnoticed. The gene that was referenced here is *mfI061*, which is now indicated between parenthesis in the main text at line 203.

Page 12 Lines 279-282 The sentence "In parallel, the original transposon mutagenesis data were reanalyzed, resulting in the re-assignment of 79 coding genes previously considered as non-essential, for a total of 332 *M. florum* genes determined as essential (Supplementary figure 11, Supplementary file 9, and Material and methods)." Needs more explanation. If a gene was disrupted by a transposon, then it cannot be essential.

We agree with the reviewer that in principle, a gene hit by a transposon should be considered dispensable. Nevertheless, Hutchison *et al.* (2016) have shown that genes with an insertion in the last 20% of their nucleotide sequence could retain their activity. This approach was used in our assessment of gene essentiality for this manuscript. While these considerations were mentioned in the Materials and Methods (now at lines 731-735) and in the Appendix Supplementary Text (Gene essentiality section), we now specify this in the main manuscript at lines 271-276 with the appropriate reference (Hutchison *et al.*, 2016):

"In parallel, the original transposon mutagenesis data were reanalyzed following the method proposed by Hutchison and colleagues (Hutchison *et al.*, 2016), where genes hit solely in the final 20% of their nucleotide sequence were not considered as essential. Using this approach resulted in the re-assignment of 79 coding genes previously considered as non-essential, for a

total of 332 *M. florum* genes now determined as essential (Figure EV3, File EV8, and Materials and Methods)."

Page 21 Line 452-455 Mentioned in their 2019 eLife paper "Essential metabolism for a minimal cell" and their 2020 BioRxiv paper "Genetic requirements for cell division in a genomically minimal cell" the JCVI-syn3A team reports a number of genes in that organism that can be removed individually although deletion of multiple seemingly unrelated genes sometimes greatly slowed growth rates. That said, I suggest rewording the sentence in this manuscript "Given that JCVI-syn3.0 was pleomorphic, had a doubling time three times greater than its parent JCVI-syn1.0, and that 19 genes had to be re-inserted to produce the more robust JCVI-syn3.0A⁸, predictions containing fewer genes than this organism are not likely to be viable."

We thank the reviewer for this very insightful reference that clearly illustrates our claim. We have therefore reformulated as suggested (lines 440-443). It now reads:

"While removing individual genes from JCVI-syn3.0 is still possible, combining multiple gene deletions often resulted in greatly reduced growth rates (Breuer *et al*, 2019; Pelletier *et al*, 2021). Hence, predictions containing fewer genes than this organism are not likely to be viable."

Reviewer #3:

In this paper, the authors present iJL208, the first genome-scale model (GEM) of metabolism for *Mesoplasma florum*, a fast-growing near-minimal organism. The model accounts for ~30% of the protein-coding genes. The authors have used computational approaches that rely on both sequence and structural homology to investigate and review the gene annotations. The three computational methods, proteome comparison, structural homology, and enzyme commission (EC) number identification, were combined to assign a final annotation score for each *M. florum* protein. A novel semi-defined growth medium was developed to identify the primary energy sources metabolized by *M. florum*. The model constraints for substrate uptake and product secretion rates were determined in the novel medium. Genome-wide expressions and essentiality datasets and the growth data on different carbohydrates were used to validate and refine iJL208. The flux states and gene essentiality predictions were made to validate the model against genome-wide expression and essentiality datasets. Finally, the authors used the phylogenetic proximity of *M. florum* with the minimal cell *Mycoplasma mycoides* JCVI-syn3.0 to assess the predictive power of GEMs for designing minimal genomes. They used an experimentally validated iJL208 model to formulate a minimal genome prediction. It can account for both transcription unit architecture and genome-wide essentiality

We wish to thank Reviewer #3 for his/her positive and constructive feedback, which overall improved our manuscript. We present here our point-by-point response in blue, with sentences from the revised manuscript in red.

Comments

Major

- Line 125, It was mentioned that the pseudo-reactions (95) were excluded while calculating the fraction of reactions in the model that are transport reactions. The denominator of the fraction of transport reactions is mentioned 278, which is not consistent if we consider a total of 370 reactions (excluding pseudo-reaction will give 275 reactions). Can the author provide the details of the 84-transport reaction and 278 reactions, if any excluded reactions are considered exceptions?

Based on this comment and to avoid any confusion, we modified the paragraph accordingly (lines 117-123):

Overall, 236 of the 370 reactions are gene-associated in iJL208 (Figure 2 and File EV4). Of those, 156 reactions are linked to a single gene while 80 are linked to more than one (enzyme complex or isozymes). Of the 134 orphan reactions, 93 are pseudo-reactions (85 extracellular exchanges, three intracellular sinks, one ATP maintenance, and four biomass reactions) while 41 are necessary orphans (15 spontaneous and 26 orphan transport reactions). Notably, about a third (84/277) of the total number of reactions (excluding pseudo-reactions) in the model are transport reactions.

- Page 6: Line 121: Fig 2, Can the author provide the details of pseudo-reactions in the figure? What is the consideration of such reactions in GEM models? Line 121 suggests 236/370 gene associated reactions, but Fig. 2 shows a count of 275 reactions. The discrepancy is due to the consideration of 39 CO₂ transport reactions in the module reactions in fig. 2. The author must include the details of transport reactions in the figure to avoid such issues. Line 128-129 suggests that the iJL208 reactions were grouped into six different modules (275 reactions). No information about pseudo-reactions is mentioned.

We replied to the previous comment by indicating the definition of every type of pseudo-reaction included in the model (lines 117-123). We have also modified Figure 2 to add the number of orphan reactions that each module contains as requested by the reviewer, and modified the figure legend accordingly. Finally, we have added a reference to the Appendix Supplementary Text and File EV4 (line 127), which detail the composition of each module in the model along with manual curation information.

- Page 7: Line 135-142: Fig. 3A, Can the author provide the proportion of orphan reactions for six modules. Fig. 3A does not suggest glycan in the list of the highest percentage of orphan relative to the module's total reaction. A quantitative description in the figure will be required to corroborate text with the figure.

The detailed number of orphan reactions per module is now presented in Figure 2. We also thank the reviewer for noticing a mistake about the glycan module. Indeed, the glycan module contains a single orphan reaction as shown in Figure 3A. However, the proportion of reactions that were assumed promiscuous is high in this module (12/15), and therefore this module is not as well defined. The main text was modified to better reflect this situation (lines 133-138):

"Conversely, the Lipids and the Vitamins & Cofactors modules had the highest percentage of orphans relative to their total number of associated reactions (25% and 33%, respectively). While the Glycans module contains a single orphan reaction, the majority of its reactions (12/15) are assumed promiscuous reactions. In proportion to their total number of genes, these three modules also displayed a lower gene annotation confidence level than the other three (Figure 3B)."

- Line 144-149: table 1 suggests that the total number of genes in the model is 680, but Line 146 mentions 685 genes. The author must correct the number of genes in the model, either in table 1 or the paragraph, to make the data consistent.

The number of genes in the model is 208 as stated in Table 1 (Model/Total). The number of predicted genes in the *M. florum* genome varies from one annotation system to another. Table 1 is based on the

RefSeq annotation system, in which 680 protein-coding genes are present. The number 685 was obtained from the PATRIC annotation, which can indeed be confusing. For consistency, we corrected the text to indicate 411/680 genes (line 141).

- Fig. 7B, The Venn diagram showed does not justify the numbers mentioned in it. Brief labeling alongside the numbers in the Venn diagram can give more insight into the data. For example, the number of deletions shown in the Venn diagram is $(37+32+76 = 145)$, which does not match with the number (152) mentioned in Line 310. And a similar case with the conserved proteins. The author must correct the data shown in the Fig. 7B. There is a missing color label for environment information processing (EIP) in Fig. 7C mentioned only in the figure caption. The author must correct the labeling issues.

We are sorry about this discrepancy between the numbers and have modified the text as follows to match with the numbers presented in Figure 7B (lines 305-307):

"The smallest genome size (562 kbp) was identified at the lowest imposed growth rate, corresponding to 563 retained and 152 deleted genes (535 and 145 protein-coding genes, respectively) (Figure 7A)."

We also modified Figure 7 legend and caption to specify the total number of retained and deleted proteins:

Figure 7B legend: "Conserved proteins shared with:" replaced by "535 retained proteins shared with:". "Deleted proteins shared with:" replaced by "145 deleted proteins shared with:".

Figure 7A caption: "Faded circles represent genome size and bright triangles the number of **coding and non-coding** deleted genes".

Figure 7B caption: "Venn diagram showing the **protein coding genes** shared between".

Since only a single protein was associated with the environment information processing (EIP) category in Fig. 7C, we corrected the Fig. 7 caption as follows to avoid unnecessary labeling (line 1100-1102):

"The single protein associated with environment information processing is represented in black on the top part of the diagram between the genetic information processing and metabolism categories."

Minor

- Fig. 1F, Line 104, the number of proteins mentioned in the result section for different confidence intervals does not look the same as shown in the plot. Can the author provide the count for each final annotation score along with the histogram?

We thank the reviewer for pointing that out. The counts were revised (line 105) and the Figure 1F was corrected accordingly, with the number of proteins for each final score now appearing directly on the histogram.

- Line 382: The author needs to describe the PTS system before using the abbreviation.

The PTS abbreviation (phosphotransferase system) is now defined at line 255 of the manuscript.

- Table 3, * and ** shows extrapolation from other species and mollicutes models, but no method was demonstrated.

Table 3 shows values used in other models available in literature; we did not generate these numbers but extracted them from these publications. To make this point more obvious to the reader, we added the reference of each model next to the species name.

- Line 518: Homology Modeling: C-score and Tm-scores were not described. The author must describe those scores and what it signifies.

These scoring systems are used by the I-TASSER reconstruction tool and are defined here: <https://zhanglab.ccmb.med.umich.edu/I-TASSER/about.html#:~:text=What%20is%20C%2Dscore%3F,of%20the%20structure%20assembly%20simulations>.

As requested by the reviewer, we added these definitions to our Materials and Methods (lines 500-503).

- Line 593, 751: No description of MCC was shown in the method section. The author must provide that.

As requested, the Matthews correlation coefficient is now mathematically defined at lines 566-568 of the Materials and Methods section (Equation 4).

- Line 619: No reference for previous work

The appropriate references are now cited at line 594 of the manuscript (Matteau et al, 2020, 2015).

- Line 659: the line specifies growth rate as k , whereas eq. 4 and 5 consider r as growth rate. No use of k is mentioned anywhere else in the article.

We thank the reviewer for pointing out this error. We changed k to r as it should be (line 630).

- Line 683: what column is XXX must be described.

We thank the reviewer for pointing out this other error. The column used for lactate and acetate quantification is now properly described in this section. We also revised the entire paragraph and added additional details regarding the HPLC quantifications. This section now reads (lines 646-656):

“A Dionex CarboPac SA10–4 μM column was used for sucrose quantification, while a Dionex IonPac AS11-HC-4 μm IC column was used for lactate and acetate quantification. The injection volume was set to 5 μl , and both the electrochemical detector and columns were operated at a temperature of 30°C. Mobile phase was composed of aqueous KOH solution and the elution gradient mode was set as follows: sucrose, 1 mM for 12 min, 10 mM for 5 min, and 1 mM for 10 min; acetate and lactate, 1 mM for 5 min, 15 mM for 9 min, and 30 mM for 11 min. The flow rate was maintained at 1.25 $\text{ml}\cdot\text{min}^{-1}$ for sucrose, and 1.5 $\text{ml}\cdot\text{min}^{-1}$ for lactate and acetate. For sucrose quantification, the stability of the signal was ensured by a 200 mM KOH post-injection using a Dionex GP 50 gradient pump set to 0.25 $\text{ml}\cdot\text{min}^{-1}$. Quantifications were performed by external calibration using 99.95% sucrose (Acros), 98% anhydrous L-lactic acid (Alfa Aesar), and 99.7% acetic acid (Fluka).”

- Line 701-702: What is the other sucrose concentration that was considered? Can the author provide the variation in the maximum substrate and product-specific rates for different sucrose concentrations? Also, give the reasoning behind considering only the two highest initial sucrose concentrations.

Based on this comment, we precised our rationale. This section now reads (lines 668-672):

“The maximum substrate and product specific rates are expected to reach a plateau as the initial concentration of substrate increases. We estimated that this plateau would be reached at the two highest initial sucrose concentrations tested in this study (0.05 and 0.1%). For modeling purposes, substrate and product specific rates were therefore determined by computing the average of the possible rates obtained for these two initial sucrose concentrations.”

- Page 34, Eqn. 7, The author must describe the parameters used in the equation. The authors must fix the formatting of the equations.

We thank the reviewer for pointing this out. Indeed, what was written did not correspond to the Hadamard product of two identical matrices that was executed to generate the Correlation Score. The equation (now Equation 8) and formulation were therefore changed and now reads (lines 717-720):

A correlation score ($S_{i,j}$) was obtained by multiplying the MCC score for each threshold pair ($M_{i,j}$) by the number of genes expressed at these same thresholds ($X_{i,j}$):

$$(Eq. 8) \quad S_{i,j} = X_{i,j} \cdot M_{i,j}$$

Thank you for sending us your revised manuscript. We have now heard back from the two reviewers who were asked to evaluate your study. As you will see, the reviewers are satisfied with the modifications made and think that the study is now suitable for publication.

Before we can formally accept your manuscript, we would ask you to address the following issues:

REFEREE REPORTS

Reviewer #2:

I think this paper is ready for publication. I have been looking forward to seeing the authors' update. This is a terrific body of work.

Reviewer #3:

Satisfied with the replies and changes to the manuscript. No additional comments.

The authors have made all requested editorial changes.

Thank you again for sending us your revised manuscript. We are now satisfied with the modifications made and I am pleased to inform you that your paper has been accepted for publication.

Corresponding Author Name: Sébastien Rodrigue

Manuscript Number: MSB-2020-10099